# A gut-brain-gut axis orchestrates host responses counteracting microbiome-induced iron insufficiency

Guanqun Li[1,2,3,4], Yangyang Wu[1,2,3,4], Xiaowen Huang ID [1,2,3,4], Minghui Du[1,2,3] & Hongyun Tang ID [1,2,3✉]

## Abstract

**The brain monitors changes in the gut microbiome to maintain health, but the impact of specific bacterial alterations, as well as the underlying mechanisms, remains largely unclear. Here, we discovered an unexpected neuronal regulation of iron metabolism, mediating the neuronal surveillance of gut bacterial activity in *C. elegans*. Specifically, through a genome-wide screen, we identified 29 *E. coli* genes, mainly linked to metabolism regulation, whose inactivation could increase dopamine and serotonin biosynthesis in *C. elegans* head neurons. These neurons respond to the lack of respiratory chain genes in *E. coli* in the gut by perceiving intestinal mitochondrial impairment resulting from bacterial-induced reduction in host labile iron levels. Such neuronal responses subsequently promote intestinal ferritin-1 expression to counteract bacterially induced labile iron reduction, thus maintaining mitochondrial function. Our findings reveal how alterations in bacterial metabolism can elevate dopamine and serotonin levels in the host brain, demonstrating that the nervous system not only senses microbiome-caused changes in the gut but also feeds back to revert them.**

**Keywords** Bacteria–Host Interaction; Dopamine; Gut–Brain–Gut Axis; Iron Metabolism; Serotonin
**Subject Categories** Metabolism; Microbiology, Virology & Host Pathogen Interaction; Neuroscience

## Introduction

Throughout the life of an animal, the activity and community composition of gut microbes change in numerous ways, which is implicated to profoundly affect the host's physiological activities (David et al, 2014; Holmes et al, 2012; Morais et al, 2021; Vich Vila et al, 2020). To maintain the health of the host, the brain has been suggested to monitor these changes in gut bacteria and orchestrate appropriate responses (Gabanyi et al, 2022; Meisel et al, 2014; Morais et al, 2021; Muller et al, 2020). Accumulating evidence, including the results from fecal microbes transplantation

experiments, supports that host nervous system responds to the changes in the gut microbial composition to adjust various physiological processes, which is important for promoting health such as relieving behavioral abnormalities and maintaining gut homeostasis (Buffington et al, 2016; Garcia-Cabrerizo et al, 2021; Morais et al, 2021; Obata et al, 2020; Sgritta et al, 2019). However, whether and how the alterations in gut microbial activity, occurring in response to various host and environmental factors, such as aging, diet and drug treatment (David et al, 2014; Morais et al, 2021; Vich Vila et al, 2020) and thereby contributing heavily to the diversity of bacterial stimulations that act on the host, can be monitored by the nervous system has barely been explored. For example, metabolic activity of gut bacteria exerts influence on host through various means such as facilitating the digestion and metabolism of the host and generating effective metabolites (Franzosa et al, 2019; Rowland et al, 2018; Valdes et al, 2018) and thus is speculated to critically impact host biological processes, but a full investigation of the potential of these microbial metabolism changes in altering activity of the nervous system is lacking. Therefore, systematically examining microbial activity change-mediated neuronal responses and elucidating the underlying mechanisms are of great importance, which should provide comprehensive insights into understanding the microbes–brain interactions.

Remarkably, gut bacteria are implicated to be able to alter the biosynthesis of neurotransmitters to profoundly affect host physiology (Hamamah et al, 2022; Morais et al, 2021; Yano et al, 2015). For example, it was reported that gut microbiota promotes serotonin production from the colonic enterochromaffin cells to affect the gastrointestinal motility in mice (Yano et al, 2015). However, whether gut bacteria can increase the biosynthesis of neurotransmitters such as serotonin and dopamine in the brain, which in turn may modulate host physiological functions to promote health, remains largely elusive (Hamamah et al, 2022; Morais et al, 2021). Especially, the particular bacterial changes that could promote the levels of the brain neurotransmitters are unclear. Moreover, the mechanism underlying the bidirectional communication between gut bacteria and brain neurons is still not well understood (Morais et al, 2021). Although several studies report the impacts of gut on the brain or brain on the gut, a bidirectional gut–brain–gut regulatory loop for any specific bacterial stimuli has not been clearly elucidated. Altogether, identifying bacterial activity changes that are capable of increasing brain dopamine and

[1]Key Laboratory of Growth Regulation and Translational Research of Zhejiang Province, School of Life Sciences, Westlake University, Hangzhou, Zhejiang, China. [2]Westlake Laboratory of Life Sciences and Biomedicine, 310024 Hangzhou, China. [3]Institute of Biology, Westlake Institute for Advanced Study, 310024 Hangzhou, China. [4]These authors contributed equally: Guanqun Li, Yangyang Wu, Xiaowen Huang. ✉E-mail: tanghongyun@westlake.edu.cn

serotonin biosynthesis, and dissecting a potential gut–brain–gut axis that may mediate the physiological roles of this activation of dopaminergic and serotonergic neurons, are highly important. The resulting findings may not only deepen understanding of how gut microbiota enhance neurological function, but also hold promise for developing strategies to treat disorders associated with reduced dopamine and serotonin levels.

Emerging evidence supports that gut, the major organ that directly interacts with bacteria, likely plays a dominant role in mediating the perceiving of the gut bacteria by the host nervous system, which could orchestrate feedback responses to modulate the gut physiology and its associated microbes (Aresti Sanz and El Aidy, 2019; Muller et al, 2020; Obata et al, 2020). It has been suggested that the brain may monitor gut bacterial changes through perceiving the metabolic state of the gut (Aresti Sanz and El Aidy, 2019; Dalile et al, 2019; Morais et al, 2021), whereas the exact mechanisms are yet to be fully understood. Iron, a growth-limiting factor in the environment, is essential to life, including animals and bacteria, and the gut microbes often compete with the host for this crucial nutrient (Das et al, 2020; Qi and Han, 2018; Wilson et al, 2016), thus mechanisms should have been evolved in the host for tightly monitoring the level of iron in the gut as well as other tissues, as both insufficient or excessive body iron causes pathological consequences (Wang and Babitt, 2019). Given the prominent role of the nervous system in coordinating different tissues/organs of the body, it is an intriguing idea to explore whether the nervous system is capable of inspecting the iron metabolism change in the peripheral tissues and mediating the important host-bacteria iron talk. Unveiling whether/how the brain perceives iron metabolic state change in the peripheral tissues, such as the gut, should deepen our understanding of interorgan regulations of the systemic iron homeostasis, which will provide insights into the mechanisms underlying the surveillance of the gut and its microbial content by the brain.

*C. elegans* is widely used to study the basic rules underlying the communications between bacteria and the host (Meisel et al, 2014; O'Donnell et al, 2020; Qi and Han, 2018; Zhang et al, 2019), which has led to seminal discoveries, such as bacteria-mediated life-span extension and microbes-induced change of host behavior (Han et al, 2017; Meisel et al, 2014), enlightening the mechanisms as well as the important physiological roles of microbes-host interactions. Here, by using *C. elegans* as a model system, we aimed to obtain an overall understanding of what alterations in bacterial activities can trigger elevation of dopamine and serotonin biosynthesis in the head neurons and to identify the underlying molecular and cellular mechanisms. As mutations in bacterial genes can result in changes in microbial activities, we administered single-deletion *E. coli* mutants from the Keio library containing knockouts of all non-essential bacterial genes to *C. elegans* and assessed responses from the head dopaminergic and serotonergic neurons by analyzing the induction of BAS-1, a shared enzyme of dopamine and serotonin biosynthesis. We found that deletions of bacterial genes involved in a variety of processes, including those encoding respiratory chain components, triggered a response from these head neurons, as measured both by increased fluorescence of a GFP-fused BAS-1 transgenic reporter and by the elevation of dopamine and serotonin levels. Furthermore, we discovered that the head dopaminergic and serotonergic neurons perceived the presence of the bacteria containing mutations in respiratory chain genes by inspecting the

resultant reduction in peripheral labile iron levels. Mechanistically, we found that there exists a gut–brain–gut axis that allows the head neurons to detect bacteria-induced decreases in labile iron levels and the consequent impairment of the intestinal mitochondria and then to orchestrate appropriate responses to prevent further decreases in available iron levels in the gut, which is important for maintaining mitochondrial function of the host. Overall, our study identified the changes in bacterial activity that provoke head dopaminergic and serotonergic neuronal response to increase dopamine and serotonin biosynthesis and uncovered an unexpected neuronal regulation of peripheral iron metabolism through a gut–brain–gut axis, which exemplifies a bidirectional communication mechanism by which the brain monitors changes in the activities of the gut and gut-associated bacteria.

# Results

## Head serotonergic and dopaminergic neurons in *C. elegans* respond to changes in gut bacterial activity, including the inactivation of *E. coli* respiratory chain genes, by increasing neuronal dopamine and serotonin biosynthesis

Throughout an animal's life, the activity of gut bacteria can change in numerous ways to potentially affect host physiology, and the nervous system may monitor these alterations in microbial activity to maintain health. To investigate what types of changes in microbial activities could provoke responses from the host nervous system, we carried out a screen to explore whether treating *C. elegans* with any of the 3985 *E. coli* mutants containing single deletions of all the non-essential genes from the Keio collection (Baba et al, 2006) could trigger responses in two sets of neurons in the head of *C. elegans*, the serotonergic and dopaminergic neurons, to increase the biosynthesis of their corresponding neurotransmitters. In this screen, we assessed the expression of BAS-1, the shared enzyme for both dopamine and serotonin biosynthesis that is expressed in these neurons (Hare and Loer, 2004) (Fig. 1A), by measuring the fluorescence intensity of GFP-fused BAS-1 driven by the native *bas-1* promoter from a previously-used *Is[bas-1p::BAS-1::GFP]* transgene (Fig. 1B) (Yuan et al, 2020), and stage-matched worms were analyzed. After three rounds of testing, single deletions of 29 bacterial genes were found to lead to an increase in BAS-1 expression in the *C. elegans* head neurons when compared to the control worms treated with the wild-type parent *E. coli* strain BW25113 (Figs. 1B and EV1A; Table EV1). The identified microbial genes encode a variety of classes of proteins, including transporters, enzymes, and membrane proteins, and GO analyses indicated that these bacterial genes were mainly involved in metabolic processes and transport, including the electron transport chain (Fig. EV1B–D). These findings indicate that the inactivation of bacterial genes involved in various biological processes triggers responses from host dopaminergic and serotonergic neurons.

Interestingly, the bacterial mutants obtained from the screen were enriched in bacterial genes encoding the cytochrome bo (3) ubiquinol oxidase subunit involved in the *E. coli* respiratory chain pathway, including *cyo-A/-B/-C/-D* (Fig. 1B,C). As an example, a bacterial mutant that strongly induced BAS-1::GFP, the *E. coli cyoB* deletion mutant (*ΔcyoB E. coli*) was used to further understand the

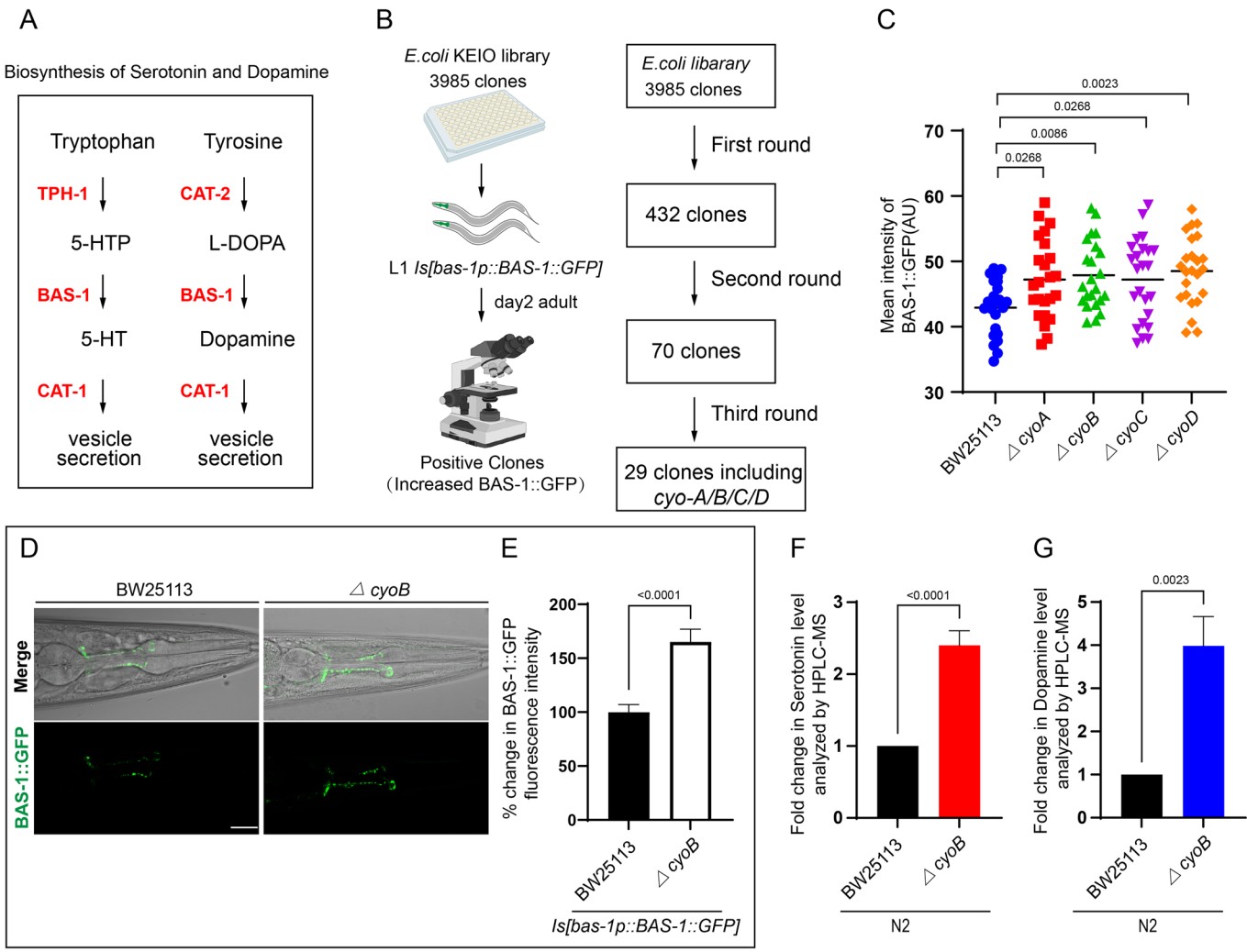

**Figure 1. Head serotonergic and dopaminergic neurons of *C. elegans* respond to bacterial activity changes, including the inactivation of *E. coli* respiratory chain genes, to increase biosynthesis of serotonin and dopamine.**

(A) Schematic diagram indicating the biosynthesis process of serotonin and dopamine neurotransmitters. 5-HTP, 5-HT and L-DOPA indicate 5-hydroxyl-L-tryptophan, serotonin and L-dihydroxyphenylanaline, respectively. TPH-1 exhibits tryptophan 5-monooxygenase activity. CAT-2 enables tyrosine 3-monooxygenase activity. BAS-1, enable L-dopa decarboxylase activity, is the shared enzyme for both dopamine and serotonin biosynthesis. CAT-1 transports dopamine and serotonin into synaptic vesicles. (B) Illustration of the genome-wide screen of the bacterial genes that enhanced *bas-1p::BAS-1::GFP* expression in the host *C. elegans* upon inactivation. The Keio library containing 3985 single deletions of all the non-essential genes of *E. coli* was used to screen for bacterial mutants that enhanced the expression of BAS-1, the shared enzyme for both dopamine and serotonin biosynthesis, as indicated in (A). The bacterial clones that significantly enhanced the expression of BAS-1::GFP in all three rounds of testing were defined as positive hits. Stage-matched worms were analyzed, and the intensity of BAS-1::GFP for each bacteria treatment was evaluated in day-2 adult stage worms. n >= 18 in each group. *P* values were determined by unpaired two-tailed *t* test. (C) Quantitative analyses of BAS-1::GFP expression induced by feeding *Is[bas-1p::BAS-1::GFP] C. elegans* bacteria with a single deletion of *cyoA/B/C/D*. Wild-type BW25113 *E. coli*, the parental strain for generating these mutations (Baba et al, 2006), was used as a negative control for analyzing the effect of these genes deletion on BAS-1::GFP expression in the *C. elegans* host. Each dot represents the mean intensity of BAS-1::GFP expression in each worm. n > 20 for each group. *P* values were determined by one-way ANOVA. (D, E) Images and bar graph indicating that *E. coli* with *cyoB* deletion (*ΔcyoB*) causes increased expression of BAS-1::GFP in *C. elegans* dopaminergic and serotonergic neurons. Wild-type worms carrying *Is[bas-1p::BAS-1::GFP]* transgene were analyzed. The bar graph shows quantification of the BAS-1::GFP fluorescence intensity which is relative to wild-type animals treated with BW25113 presented in (D). n > 30 for each group. Scale bar, 20 μm. *P* values were determined by unpaired *t* test. (F, G) Bar graph showing HPLC-MS analyses of serotonin and dopamine levels in day-2 adult N2 *C. elegans* treated with BW25113 or *ΔcyoB E. coli*. The fold change of serotonin (F) or dopamine (G) was calculated by normalizing to the levels from the *C. elegans* treated with BW25113. n > 2000 each group. *P* values were determined by unpaired *t* test. For all the panels, experiments were performed for at least three times and data are the mean ± SEM. Source data are available online for this figure.

detailed mechanism by which the head dopaminergic and serotonergic neurons respond to the bacterial activity changes (Fig. 1C). First, our further independent experiments corroborated the obvious induction of BAS-1::GFP by the *ΔcyoB E. coli* (Fig. 1D,E). qPCR analyses indicated an increase in *bas-1* mRNA

following *ΔcyoB E. coli* treatment (Fig. EV1E), supporting that transcriptional upregulation at least partially contributes to this induction of BAS-1. Next, we assess whether this increase of BAS-1 level could promote corresponding neurotransmitter biosynthesis by HPLC-MS analyses. Indeed, we found that the serotonin and

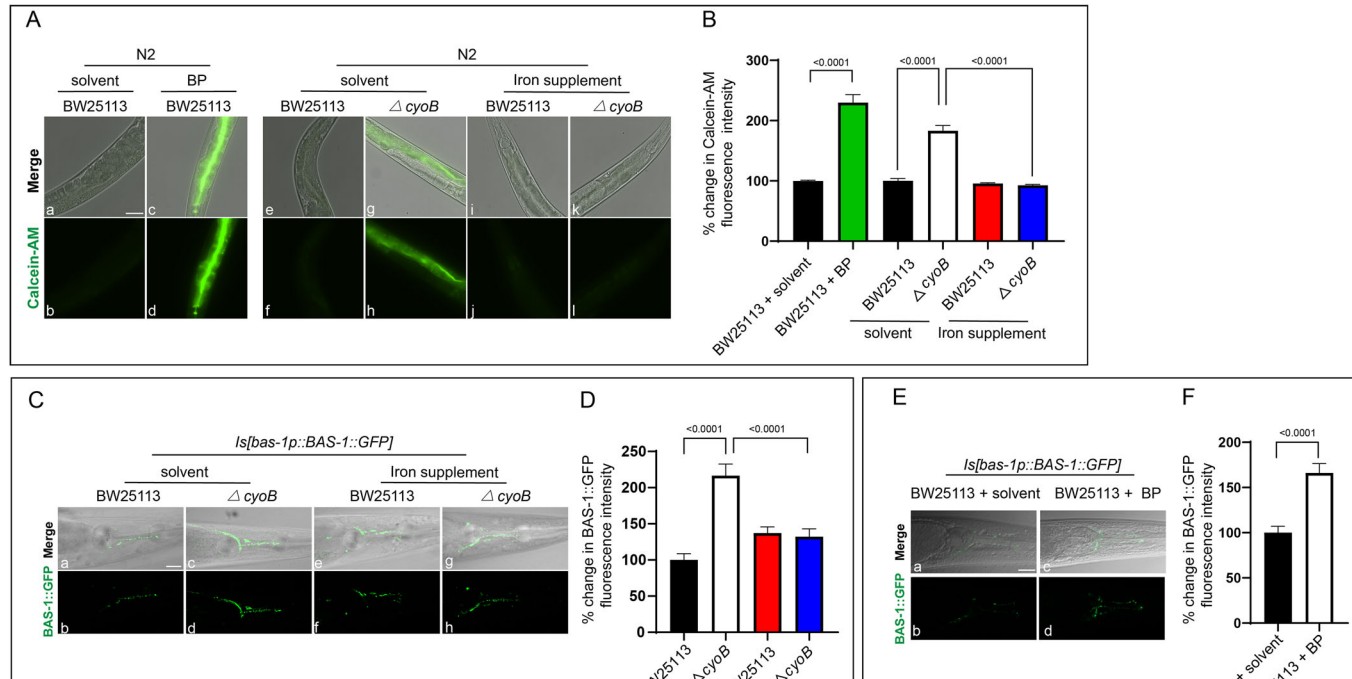

**Figure 2. Serotonergic and dopaminergic neurons respond to inactivation of the bacterial respiratory chain gene *cyoB* by detecting bacteria-caused host labile iron insufficiency**

(A, B) Images and bar graphs indicating the decrease in labile iron levels in the host *C. elegans* treated with the Δ*cyoB E. coli*. The intracellular labile iron level was measured by the widely used dye Calcein-AM, which enters cells and is immediately cleaved by intracellular esterases to form fluorescent Calcein, which is quenched by iron binding; thus, the increased fluorescence indicates a relatively low labile iron level. The iron chelator 2-2' bipyridyl (BP) caused an increase in Calcein fluorescence (c, d). The increase in Calcein fluorescence induced by the Δ*cyoB E. coli* mutant (g, h) was reversed by supplementation with iron (4 mM FeCl$_3$) (k, l). The bar graph shows percent changes in the Calcein fluorescence intensity normalized to the levels in animals treated with both BW25113 and solvent. *P* values were determined by two-way ANOVA and unpaired *t* test; *n* > 20 for each group. Scale bar, 50 μm. (C, D) Images and bar graphs showing that iron supplementation reverses the induction of BAS-1::GFP expression caused by Δ*cyoB* bacteria. The percent change of BAS-1::GFP fluorescence intensity was calculated by normalization to the levels in the animals treated with both BW25113 and solvent. Data are the mean ± SEM. *P* values were determined by two-way ANOVA. *n* > 20 for each group. Scale bar, 20 μm. (E, F) Images and bar graph depicting the induction of BAS-1::GFP expression by iron chelator 2-2' bipyridyl (BP). BP was used to sequester the labile iron. The bar graph shows the % change in BAS-1::GFP fluorescence intensity calculated relative to the levels in the solvent-treated animals. Data are the mean ± SEM. *P* values were determined by unpaired *t* test. *n* > 20 each group. Scale bar, 20 μm. Source data are available online for this figure.

dopamine levels in the wild-type worms treated with Δ*cyoB E. coli* were approximately twofold and fourfold of the levels in the animals cultured with BW25113 control, respectively (Figs. 1F,G and EV1F–K). Overall, these results demonstrate that head serotonergic and dopaminergic neurons in *C. elegans* elevate biosynthesis of corresponding neurotransmitters in response to alterations in various microbial activities, including the inhibition of *E. coli* respiratory chain genes.

## Head serotonergic and dopaminergic neurons perceive the inactivation of the bacterial respiratory *cyoB* gene by detecting bacteria-caused decrease in host labile iron level

We then aimed to understand how the dopaminergic and serotonergic neurons could perceive the presence of Δ*cyoB E. coli* to upregulate BAS-1 expression. It has been suggested that neurons could monitor the dynamics of the microbiota by detecting the bacteria themselves or by detecting bacteria-caused host changes. Since competition for iron occurs commonly between bacteria and

the host (Wilson et al, 2016; Zhang et al, 2019), we first tested whether bacteria might cause a change in iron metabolism to trigger the neuronal expression of *bas-1* in the host. To this end, we measured the cellular labile iron level, which provides available iron for metabolic demands (Kakhlon and Cabantchik, 2002), by the widely used Calcein-AM staining method. When entering the cell, Calcein-AM forms cell-impermeable fluorescent Calcein that is quenched by labile iron; thus, high Calcein fluorescence indicates relatively low labile iron levels (Kakhlon and Cabantchik, 2002; Qi and Han, 2018). 2-2' bipyridyl (BP), an iron chelator commonly used to reduce the labile iron pool in *C. elegans* (Romney et al, 2008), was used as positive control, and compared to the N2 wild-type worms treated with wild-type bacteria and solvent, the ones treated with wild-type bacteria and BP indeed displayed an increase in Calcein fluorescence (Fig. 2Aa–d,B). Interestingly, an increase in Calcein fluorescence was observed in the Δ*cyoB E. coli*-treated wild-type N2 *C. elegans* when compared to the BW25113-treated ones (Fig. 2Ae–h,B), which indicates that Δ*cyoB E. coli* causes a decrease in intracellular labile iron level in these worms. Validating that this elevated Calcein-AM signal resulted from a decrease in labile iron

rather than a potential change in cell permeability, we demonstrated that *C. elegans* exposed to *cyoB* mutant *E. coli* showed unchanged Fluo-4 AM fluorescence, a $Ca^{2+}$-sensitive dye with similar uptake and hydrolysis pathways to Calcein-AM (Fig. EV2A). Then, we confirmed this labile iron-reducing effect of the $\Delta cyoB$ *E. coli* by showing that elevating intracellular labile iron levels by iron supplementation with $Fe^{3+}$ or $Fe^{2+}$ was able to reverse the increase of Calcein fluorescence in the $\Delta cyoB$ *E. coli*-treated wild-type N2 *C. elegans* to the level observed in the N2 worms treated with wild-type bacteria (Fig. 2Ai–l,B; Appendix Fig. S1A). Furthermore, iron availability analyses were also performed by analyzing transcription level of *smf-3*, a homolog to the human divalent iron transporter DMT-1, which was known to be transcriptionally activated during iron deficiency (Romney et al, 2011). We found that, compared to BW25113-treated worms, $\Delta cyoB$ *E. coli*-treated N2 worms displayed increased *smf-3* mRNA level (Fig. EV2B), in agreement with the observation that $\Delta cyoB$ *E. coli* caused a decrease in the levels of available iron. This induction of *smf-3* transcription by $\Delta cyoB$ *E. coli* could be totally recovered by iron supplementation (Fig. EV2B), further demonstrating the labile iron-reducing effect of this bacterial strain. Altogether, these data suggest that $\Delta cyoB$ *E. coli* causes a reduction in intracellular labile iron levels in *C. elegans* host.

Next, we explored whether the observed increase in BAS-1 expression in the head neurons of worms treated with $\Delta cyoB$ *E. coli* was in response to the resultant reduction in labile iron levels. Indeed, when growing on the plates supplemented with $Fe^{3+}$ or $Fe^{2+}$, the BAS-1::GFP level from the $\Delta cyoB$ *E. coli*-treated *Is[bas-1p::BAS-1::GFP]* transgenic worms was similar to that observed in the ones treated with wild-type bacteria (Fig. 2C,D; Appendix Fig. S1B), indicating that elevating the labile iron level by the iron supplementation was able to almost completely suppress the $\Delta cyoB$ *E. coli*-induced increase in BAS-1 expression. Furthermore, we investigated whether reducing labile iron levels in *C. elegans* by an iron chelator was able to induce *bas-1* expression in the dopaminergic and serotonergic neurons. Indeed, in the *Is[bas-1p::BAS-1::GFP]* worms cultured with wild-type BW25113 *E. coli*, the expression of BAS-1::GFP was higher in the BP-treated ones than in the solvent-treated ones (Fig. 2E,F), as that observed in the $\Delta cyoB$ *E. coli*-treated worms (Fig. 1D,E). Therefore, these results indicate that labile iron insufficiency triggers a response in *bas-1*-expressing neurons. Overall, these findings indicate that the inactivation of the microbial gene *cyoB* induces a decrease in labile iron levels in the host, which is detected by dopaminergic and serotonergic neurons to upregulate *bas-1* expression.

## The increase in neuronal dopamine and serotonin biosynthesis is crucial for the *C. elegans* host to counteract the bacteria-induced reduction in labile iron levels

Having shown that serotonergic and dopaminergic neurons respond to decreased labile iron levels, we next sought to determine the function of the upregulation of BAS-1 in these neurons in response to $\Delta cyoB$ *E. coli* and to assess which neuron type mediated this response. One possibility is that these head neurons monitor bacteria-caused reductions in iron levels to orchestrate a proper response to maintain iron homeostasis. Thus, we tested whether inhibited *bas-1* expression could exacerbate the reduction in labile iron levels in $\Delta cyoB$ *E. coli*-treated worms. As predicted, in the

presence of $\Delta cyoB$ *E. coli*, the worms containing a loss-of-function (lf) mutation in *bas-1* displayed a further reduction in labile iron levels as they exhibited a dramatically higher Calcein-AM level than the $\Delta cyoB$ *E. coli*-treated N2 wild-type worms and the BW25113-treated *bas-1(lf)* worms (Fig. 3A,B), indicating the further reduction of labile iron caused by the *bas-1(lf)* in the presence of $\Delta cyoB$ *E. coli*. These results together indicate the function of elevating levels of dopamine and serotonin in the neurons for attenuating bacteria-induced reductions in labile iron levels.

To determine which of these two types of neurons mediated this host response, we tested whether *C. elegans* strains with single deletions of *tph-1* (*tph-1(lf)*) and *cat-2* (*cat-2(lf)*), which show disrupted serotonin and dopamine biosynthesis, respectively (Lints and Emmons, 1999; Sze et al, 2000), display exacerbated reductions in labile iron levels following treatment with $\Delta cyoB$ *E. coli*. Interestingly, Calcein-AM analysis indicated that both *cat-2(lf)* and *tph-1(lf)* animals displayed an exacerbated reduction in the levels of labile iron in the presence of $\Delta cyoB$ *E. coli* (Fig. 3Cg-h,D,Eg-h,F), in contrast to the $\Delta cyoB$ *E. coli*-treated N2 wild-type worms, which displayed a less severe reduction in labile iron (Fig. 3Cc-d,D,Ec-d,F), and *cat-2(lf)* and *tph-1(lf)* mutant worms treated with BW25113, which showed no obvious reduction in labile iron (Fig. 3Ce-f,D,Ee-f,F). Moreover, exposure to $\Delta cyoB$ *E. coli* resulted in a more severe reduction in labile iron levels in the *tph-1(lf);cat-2(lf)* double mutant compared to either single mutant alone (Fig. EV3A), further suggesting that both serotonergic and dopaminergic neurons function together to promote labile iron levels. Therefore, when confronting bacteria-caused reductions in labile iron levels, the responses of both the serotonergic and dopaminergic neurons are required for the host to maintain iron homeostasis.

Next, we sought to determine whether serotonergic and dopaminergic neurons modulate iron homeostasis through serotonin and dopamine signaling; thus, we performed dopamine and serotonin supplementation to analyze their potential effects on alleviating *bas-1(lf)*-caused aggravated labile iron reduction. We observed that supplementation with either serotonin or dopamine was able to almost fully reverse the reduction in iron levels in the *bas-1(lf)* mutants treated with $\Delta cyoB$ *E. coli* (Fig. 3Gg–l,H), which indicates that *bas-1*-expressing neurons function through serotonin and dopamine neurotransmitters to regulate iron metabolism. Given that increasing the bioavailability of neurotransmitters by supplementation can over-activate the corresponding signaling pathways (Loer and Kenyon, 1993), these results support that over-activating either dopamine or serotonin signaling could overcome the iron reduction caused by the inhibition of both serotonin and dopamine signaling in the *bas-1(lf)* worms, respectively. Corroborating this idea, we observed that either dopamine or serotonin supplementation could even suppress $\Delta cyoB$ *E. coli*-induced reductions in labile iron levels in the wild-type worms (Fig. 3Ga–f,H), thus overactivation of either dopamine or serotonin signaling is capable of elevating labile iron levels.

To further confirm that dopaminergic and serotonergic neurons regulate iron metabolism through dopamine and serotonin signaling, we also analyzed a potential role of *cat-1*, encoding a transporter for loading monoamines, including serotonin and dopamine into the vesicle for release (Fig. 1A) (Duerr et al, 1999), in facilitating worms to cope with bacteria-induced reduction of available iron. Indeed, we found that, in the presence of $\Delta cyoB$ *E. coli*, deleting *cat-1(lf)* caused an aggravated reduction in labile

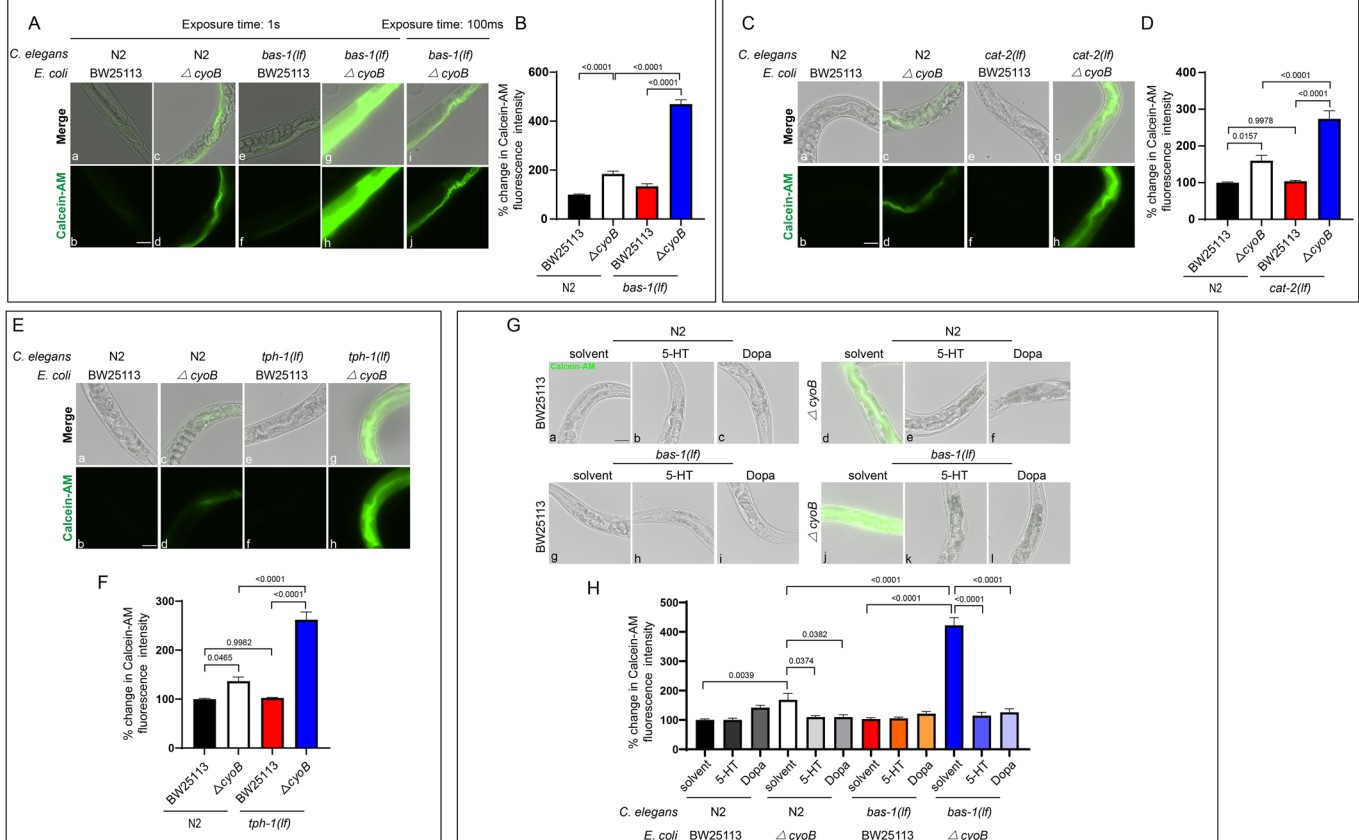

**Figure 3. The increase in *bas-1* expression in serotonergic and dopaminergic neurons in response to the presence of Δ*cyoB E. coli* is crucial for the *C. elegans* host to alleviate the resultant reduction in labile iron.**

(**A, B**) Images and bar graphs show the effects of inhibiting dopamine and serotonin biosynthesis enzyme *bas-1* on labile iron level in the presence of BW25113 and Δ*cyoB E. coli*. The percent change in Calcein fluorescence was calculated by normalization to the levels in N2 worms treated with BW25113. (**C, D**) Images and bar graph showing that deleting the dopamine biosynthesis enzyme *cat-2* aggravates the reduction in labile iron caused by Δ*cyoB E. coli*. Calcein-AM was used to indicate the labile iron, and the % change in Calcein fluorescence was calculated by normalization to the levels in N2 worms treated with BW25113. (**E, F**) Images and bar graph illustrating the effect of *tph-1* deletion on the reduction in labile iron levels at the presence of the indicated bacteria. TPH-1 is essential for serotonin biosynthesis, as shown in Fig. 1A. The percent of change in Calcein fluorescence intensity relative to the levels in N2 worms treated with BW25113 was quantified. (**G, H**) Images and bar graph indicating that the supplementation of 12.5 mg/mL dopamine (Dopa) and 10 mg/mL serotonin (5-HT) is able to reverse the Δ*cyoB E. coli*-caused free iron reduction in the wild-type N2 worms, as well as the exacerbated labile iron reduction in the *bas-1(lf)* animals treated with Δ*cyoB E. coli*. The % change in the Calcein fluorescence intensity relative to the levels in the wild-type N2 worms treated with both BW25113 and solvent is shown in the bar graph. For all the panels, experiments were performed at least three times and the data in the bar graph are the mean ± SEM. *P* values were determined by one-way ANOVA; Scale bar, 50 μm. *n* > 25 each group. Source data are available online for this figure.

iron levels (Fig. EV3B,C), when compared to the labile iron levels in the BW25113-treated *cat-1(lf)* worms and the Δ*cyoB E. coli*-treated N2 wild-type worms. In the presence of the Δ*cyoB E. coli*, the *cat-1(lf)* mutation did not exacerbate the reduction in labile iron caused by the *bas-1(lf)* mutation (Fig. EV3D), consistent with both genes acting via serotonin and dopamine to mediate iron regulation. Notably, the *cat-1(lf)* mutant exhibited a less pronounced decrease in labile iron compared to the *bas-1(lf)* mutant when exposed to Δ*cyoB E. coli* (Fig. EV3D), which may be attributed to CAT-1's role in transporting not only dopamine and serotonin but also other monoamines, such as tyramine and octopamine, which negatively regulate labile iron levels (Appendix Fig. S2). In addition, further supporting the function of dopamine and serotonin in promoting labile iron levels, the mutation of *cat-4*, which encodes GTP cyclohydrolase I, an enzyme involving in dopamine and serotonin biosynthesis (Loer et al, 2015), also significantly reduced labile iron levels in the presence of Δ*cyoB E. coli* (Fig. EV3E). Interestingly, the

*cat-4(lf)* mutant also displayed a significant reduction in labile iron when treated with BW25113 (Fig. EV3E), which may be due to CAT-4's involvement in other processes required for maintaining labile iron levels. In summary, dopaminergic and serotonergic neurons inspect bacteria-induced reductions in labile iron levels and then orchestrate the host response through dopamine and serotonin signaling to counteract this decrease.

## Head serotonergic and dopaminergic neurons respond to the bacteria-induced labile iron reduction by perceiving the consequent impairment of the intestinal mitochondria, and this neuronal response is crucial for the host to maintain mitochondrial function when confronting these bacteria

We next aimed to dissect how dopaminergic and serotonergic neurons perceive changes in labile iron levels by determining

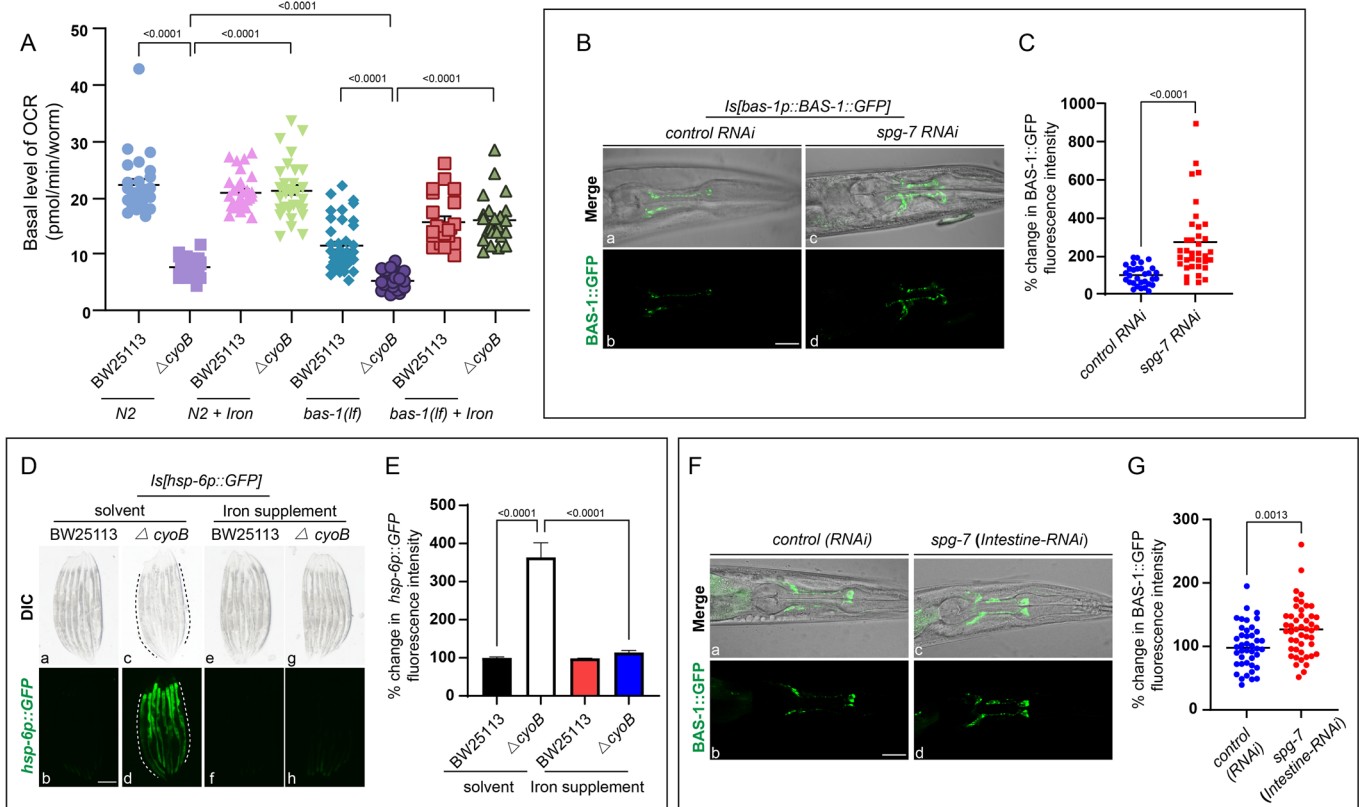

**Figure 4. Serotonergic and dopaminergic neurons respond to ΔcyoB E. coli-induced reductions in labile iron levels by perceiving the impairment of intestinal mitochondria, and this neuronal response is crucial for coping with the bacteria-caused reduction in mitochondrial function.**

(A) Graphs showing the reduction in the basal oxygen consumption rate (OCR) in wild-type animals treated with ΔcyoB E. coli and the further reduction in the OCR in dopamine- and serotonin-deficient animals incubated with ΔcyoB E. coli. Iron supplementation was performed by adding 4 mM FeCl₃ to the worm culture medium. The OCR was analyzed by a Seahorse XFe96 Analyzer as previously shown (Koopman et al, 2016), and each point indicates the OCR normalized to each worm. P values were determined by unpaired *t* test; *n* > 100 for each treatment. (B, C) Representative images and plotted graph indicating that the decline in mitochondrial function caused by *spg-7(RNAi)* is able to trigger the expression of BAS-1::GFP in head neurons. Each dot represents the % change in BAS-1::GFP calculated by normalization to the levels in worms treated with control RNAi. P values were determined by unpaired *t* test. *n* > 30 each group. Scale bar, 20 μm. (D, E) Microscopic images and bar graph showing the effects of the ΔcyoB bacteria on mitochondrial damage with or without iron supplementation. The induction of the expression of *hsp-6p::GFP*, a widely used mitochondrial stress reporter, was employed to indicate mitochondrial damage. Dashed lines indicate the intestine, and 4 mM FeCl₃ was added to the worms carrying *hsp-6p::GFP* for iron supplementation. The % change in *hsp-6p::GFP* was calculated by normalization to the levels in worms treated with BW25113. Data are the mean ± SEM. P values were determined by two-way ANOVA; *n* > 20 each group. Scale bar, 200 μm. (F, G) Microscopic images and graph indicating that *spg-7(intestinal RNAi)* is able to elevate BAS-1::GFP expression. The *rde-1(lf); kbIs7 [nhx-2p::RDE-1+rol-6(su1006)]; Is[bas-1p::BAS-1::GFP]* worm strain was used to perform *spg-7(intestine-RNAi)* for inducing intestinal mitochondrial impairment, and then analyze the resultant impact on the BAS-1 expression. The animals treated with the HT115 E. coli strain carrying the empty PL4440 vector served as the RNAi control to calculate the percent change of BAS-1::GFP fluorescence intensity. Each dot represents the percentage change of BAS-1::GFP intensity of each worm relative to the average BAS-1::GFP intensity in *control RNAi*-treated worms. P values were determined by unpaired *t* test. *n* > 30 each group. Scale bar, 20 μm. Source data are available online for this figure.

whether this process involved mitochondria, given that cellular respiration in mitochondria is one of the major processes that utilizes iron (Wang and Babitt, 2019). Thus, we tested the possibility that these neurons perceive the reduction in labile iron levels by monitoring mitochondrial impairment. First, we measured the oxygen consumption rate (OCR) to assess the change in mitochondrial function after bacterial insult. Our analyses showed that the OCR was lower in the wild-type worms treated with ΔcyoB E. coli than in those treated with BW25113 by almost twofold (Fig. 4A). Interestingly, this reduction in OCR caused by ΔcyoB E. coli could be reversed by iron supplementation (Fig. 4A). Therefore, ΔcyoB E. coli-induced labile iron insufficiency causes a decline in mitochondrial function. To probe a possible role of the mitochondrial decline in triggering response in the dopaminergic and

serotonergic neurons, we next induced mitochondrial impairment by inhibiting the function of SPG-7 mitochondrial metalloprotease that has been widely used to induce mitochondrial stress (Shao et al, 2016) and evaluated a potential induction of the BAS-1 expression in neurons. Strikingly, the increase in BAS-1::GFP expression was induced by *spg-7(RNAi)* (Fig. 4B,C), which indicates a causal role of the decline in mitochondrial function in triggering the response of dopaminergic and serotonergic neurons to the ΔcyoB E. coli-induced decrease in iron levels. Therefore, the dopaminergic and serotonergic neurons respond to the ΔcyoB E. coli-caused iron decrease by perceiving the consequent mitochondrial impairment.

We then aimed to determine whether the dopaminergic and serotonergic neurons perceive the decline in mitochondrial

function in peripheral tissue or in neurons. First, we analyzed in which *C. elegans* tissue the mitochondrial damage was caused by the Δ*cyoB E. coli* by evaluating the expression of the commonly-used mitochondrial stress reporter Is[*hsp-6p::GFP*], which can be induced in neurons as well as peripheral tissues when mitochondria are impaired in corresponding cells (Shao et al, 2016). Intriguingly, the obvious induction of *hsp-6p::GFP* expression in the intestinal cells, but not in the neurons, was observed in the Δ*cyoB E. coli*-treated worms (Figs. 4D,E and EV4A). Moreover, this induction of *hsp-6p::GFP* in Δ*cyoB E. coli*-treated worms could be suppressed by iron supplementation (Fig. 4D,E). These results indicate that Δ*cyoB E. coli*-induced decrease in labile iron levels resulted in mitochondrial impairment mainly in the intestine. Next, we tested the possibility that head dopaminergic and serotonergic neurons may sense the reduction in iron levels by perceiving this decline in mitochondrial function in the gut. To this end, we determined whether inducing mitochondrial impairment specifically in the gut with intestine-specific RNAi against *spg-7* could increase BAS-1 expression in neurons. We performed *spg-7* RNAi in the Is[*bas-1p::BAS-1::GFP*]; *rde-1(lf)*; *kbIs7* [*nhx-2p::RDE-1+rol-6(su1006)*] worms, in which the RNAi is only effective in the intestine (Espelt et al, 2005). Indeed, we found that *spg-7 (intestine-RNAi)* led to significantly higher BAS-1 expression than the control RNAi (Fig. 4F,G), indicating that BAS-1-expressing neurons monitor intestinal mitochondrial function state. Moreover, reducing labile iron levels by BP treatment caused a decline in mitochondrial function in the gut, as evidenced by the expression of *hsp-6p::GFP* in the intestine (Fig. EV4B); additionally, expression of BAS-1::GFP was induced by BP as described above (Fig. 2E,F), which further supports the idea that reduced labile iron levels caused a decline in intestinal mitochondrial function and triggered BAS-1 expression in dopaminergic and serotonergic neurons. In summary, host neurons perceive Δ*cyoB E. coli*-induced labile iron insufficiency by detecting intestinal mitochondrial impairment to trigger neuronal BAS-1 expression to orchestrate a response to recover the labile iron level.

Based on these data, we hypothesized that the response that enabled these neurons to attenuate the reduction in labile iron levels might be important for mitigating Δ*cyoB E. coli*-caused impairment of mitochondrial function. Consistent with our prediction, *bas-1(lf)* animals treated with Δ*cyoB E. coli* displayed even lower OCR than the *bas-1(lf)* animals treated with BW25113 and the wild-type worms cultured with Δ*cyoB E. coli* (Fig. 4A), and serotonin supplementation could partially reverse this decline in OCR in both wild-type and *bas-1(lf)* worms treated with Δ*cyoB E. coli* (Fig. EV4C). These data indicate that the neuronal response mediated by monoamine signaling is important for maintaining mitochondrial function when the Δ*cyoB E. coli* are present. Notably, the partial restoration of OCR in wild-type worms treated with Δ*cyoB E. coli* following serotonin supplementation (Fig. EV4C) supports the existence of a monoamine signaling-independent pathway regulating mitochondrial function. Furthermore, the further reduction in OCR in the *bas-1(lf)* worms treated with Δ*cyoB E. coli* was suppressed by iron supplementation (Fig. 4A). Together with the observation that the *bas-1(lf)* mutation aggravated the Δ*cyoB E. coli*-induced reduction in labile iron levels, these results indicate that in response to Δ*cyoB E. coli*, the upregulation of BAS-1 expression to increase dopamine and serotonin levels in these neurons plays a role in improving

mitochondrial function at least partially by promoting available iron levels.

To further investigate the important role of the induction of *bas-1* expression in neurons in maintaining mitochondrial function under conditions of iron scarcity, we tested the possibility that inhibiting *bas-1* expression would impair the ability of Δ*cyoB E. coli*-treated worms to deal with mitochondrial stress. Paraquat, a widely used mitochondrial stressor (Schaar et al, 2015), was employed to induce mitochondrial dysfunction. After paraquat treatment, the survival time of wild-type worms cultured with Δ*cyoB E. coli* was lower than that of worms treated with BW25113, which is in agreement with the decrease in the labile iron level and decline of mitochondrial function in these wild-type worms treated with Δ*cyoB E. coli* (Fig. EV4D). Importantly, this survival rate reduction could be fully reversed by iron supplementation (Fig. EV4D), which is in agreement with the mitochondrial function decline caused by the Δ*cyoB E. coli*-induced reduction in labile iron levels. These results indicate that Δ*cyoB E. coli*-induced reductions in labile iron levels impair the host's ability to cope with mitochondrial stress. Furthermore, consistent with our prediction, paraquat treatment induced an even lower survival rate in the *bas-1(lf)* mutants treated with Δ*cyoB E. coli* than in the wild-type worms treated with Δ*cyoB E. coli* or the *bas-1(lf)* mutants cultured with BW25113 wild-type *E. coli*. This *bas-1(lf)*-induced further decrease in the survival rate could be strongly reversed by iron supplementation (Fig. EV4D). Altogether, these data suggest that, in the presence of bacteria that induce reductions in available iron levels, triggering a response from dopaminergic and serotonergic neurons to recover labile iron levels is crucial for improving animals' ability to maintain mitochondrial homeostasis.

## The mitochondrial stress regulator ATFS-1 acts in the intestine to mediate dopaminergic and serotonergic neuron perception of Δ*cyoB E. coli*-induced reductions in labile iron levels

In order to understand how neurons perceive the intestinal mitochondrial impairment caused by bacteria-induced iron insufficiency, we next aimed to dissect the molecular mechanisms underlying this gut–brain communication. As ATFS-1, expressed in the majority of cells and tissues, including the intestine in *C. elegans* (Haynes et al, 2010), is known to be a master regulator for sensing mitochondrial dysfunction (Nargund et al, 2012), we thus evaluated whether intestinal ATFS-1 might be required for transmitting intestinal mitochondrial damage signals to *bas-1*-expressing neurons. We performed *atfs-1* RNAi in the Is[*bas-1p::BAS-1::GFP*]; *rde-1(lf)*; *kbIs7* [*nhx-2p::RDE-1+rol-6(su1006)*] worms for specifically knocking down *atfs-1* in the intestine (Espelt et al, 2005). Indeed, we found that, the higher BAS-1::GFP in the worms treated with the Δ*cyoB E. coli* than the ones treated with the wild-type bacteria treatment (Fig. 5A,Ba–d,C), was suppressed by the intestine-specific knockdown of *atfs-1* when compared to the treatment with the RNAi control (Fig. 5Bc-d,g-h,C), which indicates that intestinal ATFS-1 is required for the response of dopaminergic and serotonergic neurons to the mitochondrial impairment in the gut. Furthermore, we tested whether activating ATFS-1 in the intestine via a transgene expressing the dominant-active form of ATFS-1 with intestine-specific promoter *gly-19* (Nargund et al, 2012; Shao et al, 2016) was able to induce Is[*bas-1p::BAS-1::GFP*] in the presence of wild-type bacteria. Strikingly,

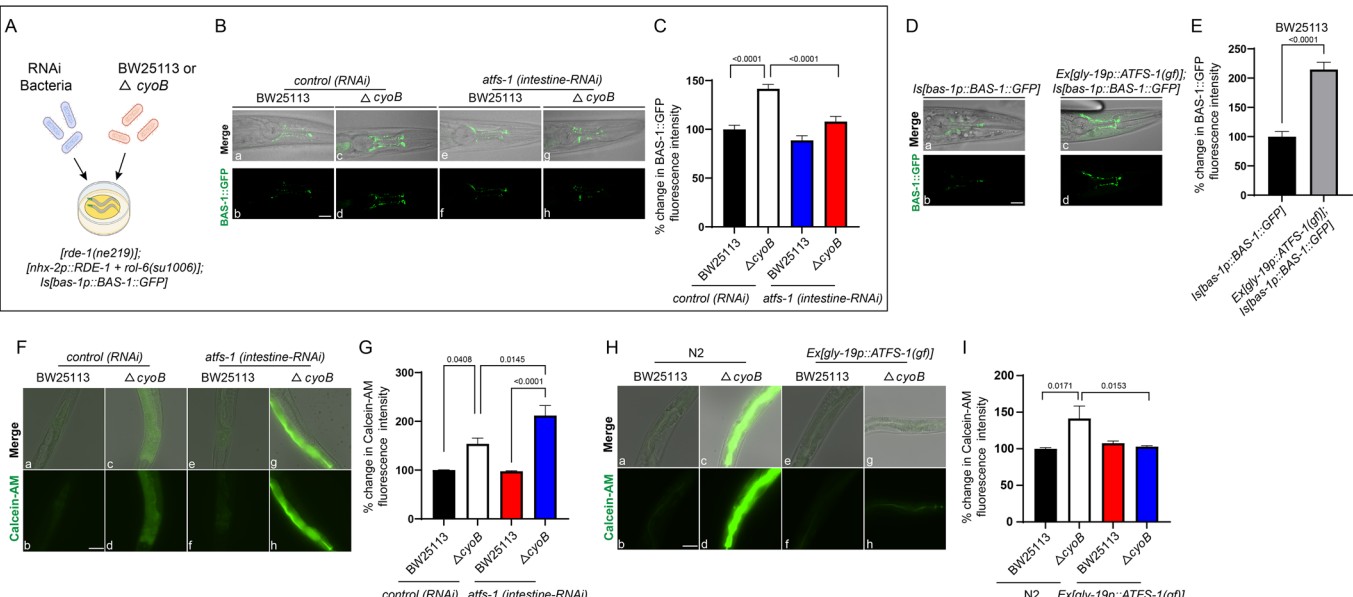

**Figure 5. The intestine-expressed mitochondrial stress sensor ATFS-1 mediates the response of dopaminergic/serotonergic neurons to ΔcyoB E. coli to attenuate reductions in labile iron levels.**

(A–C) Images and bar graphs showing that intestinal ATFS-1 is required for the ΔcyoB E. coli-induced increase in neuronal bas-1 expression. BW25113 or ΔcyoB E. coli was mixed with atfs-1 RNAi-containing bacteria in a ratio of 1:1 to analyze the role of atfs-1 in mediating the bacteria-triggered response, as described in (A). The rde-1(lf); kbIs7 [nhx-2p::RDE-1+rol-6(su1006)] worm strain expressing BAS-1::GFP underwent knock down of atfs-1 specifically in the intestine. The % change in BAS-1::GFP fluorescence was calculated by normalization to the levels in animals treated with control RNAi and BW25113. Statistics analyzed by one-way ANOVA; scale bar, 20 μm. n > 50 each group. (D, E) Images and bar graph showing that overexpression of the dominant active form of ATFS-1 in the intestine induces neuronal expression of BAS-1::GFP. The intestine-specific promoter gly-19 was used to drive the intestinal expression of atfs-1(gf) (Shao et al, 2016). The worms with indicated genotypes were treated with BW25113. The % change in BAS-1::GFP fluorescence intensity was calculated by normalization to the levels in the worms without the transgene. Statistics analyzed by unpaired t test. Scale bar, 20 μm. n > 30 in each group. (F, G) Images and bar graph indicating that inhibiting intestinal atfs-1 expression exacerbates the reduction in labile iron caused by ΔcyoB bacteria. The rde-1(lf); kbIs7 [nhx-2p::RDE-1+rol-6(su1006)] worm was used for atfs-1 intestine-specific RNAi analyses. As performed in (A), the RNAi bacteria were mixed with the BW25113 or ΔcyoB E. coli in a ratio of 1:1 to analyze the role of intestinal atfs-1 in coping with bacteria-caused reductions in iron levels. Calcein-AM analyses were used to indicate the level of labile iron. The % change in Calcein fluorescence was calculated by normalization to the levels in animals treated with control RNAi and BW25113. Statistics analyzed by one-way ANOVA; n > 20 in each group. Scale bar, 50 μm. (H, I) Images and bar graphs showing that intestine-specific overexpression of the dominant active form of ATFS-1 is able to recover the ΔcyoB bacteria-caused labile iron reduction in the N2 wild-type worms. The gly-19 promoter was used to drive the expression of atfs-1(gf) specifically in the intestine. The % change of Calcein fluorescence was determined by normalization to the BW25113-treated N2 wild-type worms. Statistics analyzed by one-way ANOVA; n > 20 in each group. Scale bar, 50 μm. Fall the panels, experiments were performed at least three times. Data are the mean ± SEM. Source data are available online for this figure.

compared to the no transgene control treatment, expression of this atfs-1(intestine-gf) transgene was able to strongly increase neuronal BAS-1 expression (Fig. 5D,E). Altogether, these data indicate that ATFS-1 acts in the intestine to perceive intestinal mitochondrial impairment and then triggers BAS-1 expression in neurons, thus mediating the impact of ΔcyoB E. coli on dopaminergic and serotonergic neurons.

Furthermore, we aimed to investigate whether intestinal ATFS-1-mediated induction of neuronal BAS-1 is involved in orchestrating response to modulate iron metabolism as the neuronal responses triggered by the ΔcyoB E. coli bacteria. First, we analyzed whether knockdown of atfs-1 specifically in the intestine aggravated iron reduction in the presence of ΔcyoB E. coli. Indeed, atfs-1(intestinal RNAi) worms treated with ΔcyoB E. coli displayed even greater reduction in labile iron levels than the atfs-1(intestinal RNAi) worms treated with wild-type E. coli or the animals treated with both control RNAi and ΔcyoB E. coli (Fig. 5F,G), which was similar to the findings observed in the bas-1(lf) group (Fig. 3A,B). Conversely, intestine-specific expression of the dominant-active form of ATFS-1 was able to fully suppress the ΔcyoB E. coli-induced reduction in labile iron levels

in the wild-type N2 worms (Fig. 5H,I), which was also observed in the atfs-1(gf) mutant worms (Fig. EV5A,B), indicating that ATFS-1 mainly functions in the intestine to cope with ΔcyoB E. coli-caused labile iron reduction. In summary, intestinal atfs-1 senses the iron insufficiency-induced impairment of intestinal mitochondria to promote BAS-1 expression in dopaminergic and serotonergic neurons to in turn alleviate intestinal iron deficiency, thus mediating the perception of ΔcyoB E. coli by the dopaminergic and serotonergic neurons.

## Neuronal CWN-2 Wnt mediates the role of intestinal ATFS-1 in elevating bas-1 expression in dopaminergic and serotonergic neurons to increase labile iron levels

Our data suggested that cell non-autonomous signaling may mediate communication between the intestinal cells and bas-1-expressing neurons to modulate iron metabolism. Discovering the cell non-autonomous signal that regulates iron metabolism is highly important, given that maintaining systemic iron homeostasis is critical for organism health. Thus, we next explored whether any endocrine signals might be involved in regulating ΔcyoB E. coli-

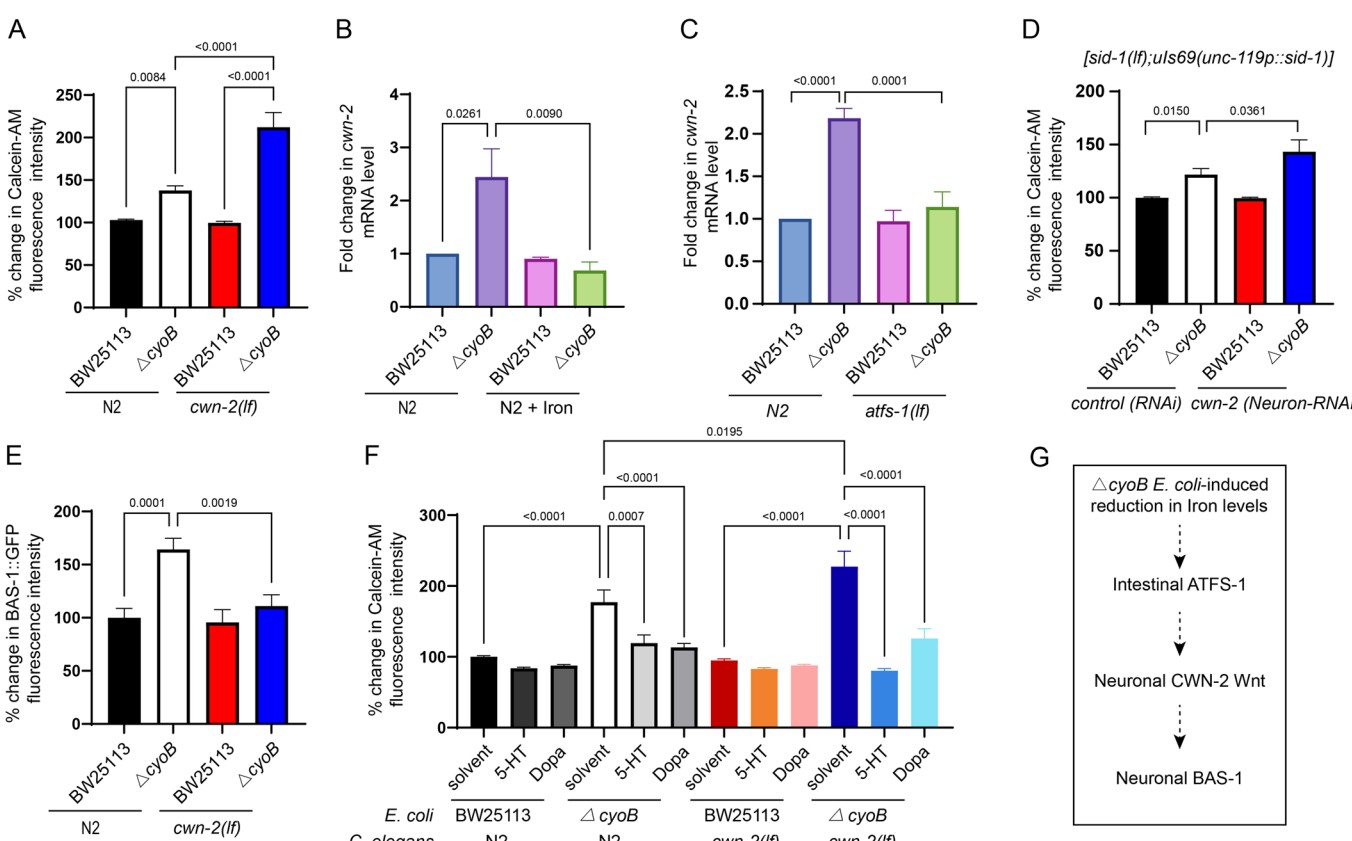

**Figure 6. Neuron-expressed CWN-2 Wnt acts downstream of ATFS-1 to mediate the response of dopaminergic/serotonergic neurons to ΔcyoB E. coli-caused labile iron insufficiency.**

(A) Bar graph showing the effect of CWN-2 Wnt ligand mutation on ΔcyoB E. coli-caused reduction in labile iron levels analyzed by Calcein-AM staining. The data are normalized to Calcein fluorescence in wild-type N2 C. elegans treated with BW25113 bacteria. n > 50 in each group. (B) qPCR measurements of cwn-2 mRNA levels indicating that ΔcyoB E. coli bacteria-caused labile iron insufficiency induced the transcription of cwn-2 in C. elegans. 4 mM FeCl₃ was used for iron supplementation. Fold change in cwn-2 mRNA levels was analyzed by normalizing to the levels in BW25113-treated animals. n > 200 each group. (C) qPCR measurements of cwn-2 mRNA levels showing that the increase in cwn-2 transcription caused by ΔcyoB E. coli is dependent on atfs-1. The fold change in the levels in animals subjected to each treatment was calculated by normalizing to the levels in wild-type animals treated with BW25113 bacteria. n > 200 each group. (D) Bar graph showing that knocking down cwn-2 in neurons exacerbates the ΔcyoB E. coli-induced reduction in labile iron in the C. elegans host. The neuron-specific RNAi strain TU3401[sid-1(lf);uIs69(unc-119p::sid-1)] was employed to knock down neuronal cwn-2. n > 20 for each group. (E) Bar graph indicating that cwn-2 is required for the ΔcyoB E. coli-triggered increase in BAS-1 expression. Is[bas-1p::BAS-1::GFP] and cwn-2(lf);Is[bas-1p::BAS-1::GFP] animals were treated with the indicated bacteria. The fold change in BAS-1::GFP fluorescence intensity was calculated by normalization to the Is[bas-1p::BAS-1::GFP] animals treated with BW25113. n > 30 for each treatment. (F) Bar graph indicating that cwn-2 functions upstream of dopamine and serotonin signaling to attenuate bacteria-caused iron reduction analyzed with Calcein-AM staining. Supplementation with 12.5 mg/mL dopamine (Dopa) or 10 mg/mL serotonin (5-HT) was able to recover the labile iron level in N2 and cwn-2(lf) worms treated with ΔcyoB E. coli. An equivalent volume of water was used as the solvent control. n > 30 for each treatment. (G) A schematic depicting the functional relationship of the indicated proteins in the process through which dopaminergic and serotonergic neurons perceive the presence of ΔcyoB E. coli. For all the panels, experiments were performed for at least three times, and the data are the mean ± SEM. P values were determined by one-way ANOVA. Source data are available online for this figure.

induced changes in iron metabolism, and genes encoding ligands and receptors that are involved in several major signaling pathways were tested by analyzing the impact of their deletions on labile iron level in the presence of ΔcyoB E. coli (Figs. 6A and EV6A–F). Interestingly, the labile iron reduction was observed to be more pronounced in cwn-2(lf) worms treated with ΔcyoB E. coli compared to cwn-2(lf) worms treated with wild-type E. coli or N2 wild-type worms treated with ΔcyoB E. coli (Fig. 6A). Therefore, deletion of the CWN-2 Wnt ligand exacerbated the reduction in labile iron levels caused by ΔcyoB E. coli, which was similar to the effect observed in the bas-1(lf) mutant worms (Fig. 3A,B), suggesting a potential increase of CWN-2 in response to ΔcyoB E. coli for alleviating the iron insufficiency. Thus, we next analyzed

the transcription of cwn-2 by qPCR in the N2 worms treated with ΔcyoB E. coli and BW25113 wild-type bacteria. As predicted, we found that the transcription level of cwn-2 was increased by ΔcyoB E. coli treatment (Fig. 6B), and this increase was entirely suppressed by iron supplementation (Fig. 6B). These results together indicate that cwn-2 levels are increased in response to labile iron insufficiency caused by ΔcyoB E. coli to counteract further reductions in iron levels.

Next, we investigated the functional relationship between cwn-2 and atfs-1. We found that the ΔcyoB E. coli-induced increase in cwn-2 transcription was suppressed by the atfs-1(lf) mutation (Fig. 6C), indicating that CWN-2 acts downstream of ATFS-1 to respond to ΔcyoB E. coli-induced labile iron insufficiency. Given

the obvious neuronal expression of *cwn-2* (Kennerdell et al, 2009), we then performed tissue-specific RNAi analyses to investigate whether the CWN-2 Wnt ligand originates from neurons to act downstream of ATFS-1 to modulate iron metabolism. Indeed, when performing the neuron-specific RNAi against *cwn-2* by using the *sid-1(lf);uIs69(unc-119p::sid-1)* strain, a further reduction in labile iron levels occurred in the presence of Δ*cyoB E. coli* (Fig. 6D). The less severe reduction in the labile iron in the *cwn-2(Neuron-RNAi)* worms treated with Δ*cyoB E. coli* than in the *cwn-2(lf)* worms treated with Δ*cyoB E. coli* may be due to the partial knockdown of *cwn-2* by RNAi. These results support that the neuron-originated CWN-2 Wnt ligand, which responds to the intestinal ATFS-1, is required for the host to orchestrate a response to Δ*cyoB E. coli* challenge to attenuate reductions in available iron levels.

In addition, in our screen shown in Fig. EV6A–F, we found that *egl-20*, another Wnt ligand, was also required for alleviating Δ*cyoB E. coli*-induced iron reduction (Fig. EV6A), confirming the unexpected role of Wnt signaling in modulating iron metabolism. Interestingly, from our screen, we found that inhibiting the insulin pathway was able to recover the labile iron level in worms treated with Δ*cyoB E. coli* (Fig. EV6E,F), suggesting a repressing role of insulin signaling in regulating labile iron level. It is worth testing the role of the insulin pathway in mediating bacteria–host iron talk in the future. Altogether, our data support that Wnt signaling originating in neurons acts downstream of intestinal ATFS-1 to mediate the host response to Δ*cyoB E. coli* for counteracting bacteria-induced reductions in labile iron levels.

Our data above indicated that both *bas-1* and *cwn-2* were expressed in neurons to positively regulate labile iron levels. Thus, we next evaluated the functional relationship between CWN-2 and serotonin and dopamine signaling. We found that the Δ*cyoB E. coli*-induced expression of BAS-1 was repressed by the deletion of *cwn-2* (Fig. 6E). In contrast, the *bas-1(lf)* mutation showed no obvious effect on *cwn-2* expression induced by Δ*cyoB E. coli* (Fig. EV6G). These results indicate that *cwn-2* acts upstream of *bas-1* and downstream of intestinal ATFS-1 to mediate the dopaminergic and serotonergic neuron response to the presence of Δ*cyoB E. coli*. Furthermore, overactivation of dopaminergic and serotonergic signaling by supplementing dopamine and serotonin, respectively, not only strongly suppressed the reduction in iron levels in the wild-type worms cultured with Δ*cyoB E. coli* but also almost entirely reversed the *cwn-2(lf)*-induced further reduction in labile iron levels in the presence of Δ*cyoB E. coli* (Fig. 6F). This result further demonstrates that BAS-1-involved dopamine and serotonin biosynthesis act to mediate the role of CWN-2 Wnt in promoting labile iron level. Overall, in the presence of Δ*cyoB E. coli, bas-1* expression in dopaminergic and serotonergic neurons is triggered by a *cwn-2* Wnt signal originating in neurons to elevate peripheral labile iron levels for alleviating the bacteria-induced reductions in iron levels (Fig. 6G).

### The increase in BAS-1 expression in head dopaminergic and serotonergic neurons attenuates bacteria-induced reduction in labile iron levels by promoting intestinal *ferritin-1(ftn-1)* expression, which indicates an unexpected gut–brain–gut axis that enables the host to cope with bacteria-caused reductions in available iron

Next, we aimed to investigate the molecular mechanism by which the induction of *bas-1* expression in dopaminergic and serotonergic neurons attenuates the reduction in labile iron levels. Since cellular labile iron levels can be elevated by increasing iron absorption, we first tested whether the gene expression of the iron transporters SMF-1/-2/-3 for absorbing iron is mediated by neuronal BAS-1 after bacteria treatment. Our results indicated that compared to the wild-type bacteria control, the Δ*cyoB E. coli* treatment increased the transcription of *smf-1/-2/-3* (Figs. EV2B and EV7A–C); however, we found that deletion of *bas-1* did not affect the induction of *smf-1/-2/-3* expression by Δ*cyoB E. coli* treatment (Fig. EV7A–C), indicating that the Δ*cyoB E. coli*-induced expression of *smf-1/-2/-3* is not dependent on the activation of *bas-1*-expressing neurons. Therefore, in the presence of Δ*cyoB E. coli*, the dopaminergic and serotonergic neurons are unlikely to elevate iron levels by increasing the expression of these iron transporters.

Iron storage is known to repress labile iron levels through iron sequestration by ferritin, and ferritin levels have been previously shown to be reduced in response to iron deficiency to release labile iron (Romney et al, 2008). We thus tested whether dopaminergic and serotonergic neurons might elevate the free iron level by decreasing iron storage protein ferritin, a critical iron storage factor whose expression was reported to be repressed at iron-scarce conditions (Romney et al, 2008). Surprisingly, our LC-MS/MS analyses indicated that, compared to the BW25113 parent bacteria control treatment, Δ*cyoB E. coli*, which displayed a strong iron-reducing effect, increased rather than decreased FTN-1 protein level (Fig. EV7D), whereas FTN-2 levels remained unchanged (Fig. EV7E). Further qPCR analyses showed that, compared to the BW25113 parent bacteria control treatment, Δ*cyoB E. coli* significantly increased *ftn-1* transcription (Fig. 7A), supporting that the increase in FTN-1 protein is regulated at the transcriptional level. Furthermore, analysis of a *ftn-1* transcriptional reporter (*ftn-1p::GFP*) confirmed this obvious increase in *ftn-1* transcription in response to the presence of Δ*cyoB E. coli* (Fig. 7B,C). Specifically, we observed that *ftn-1* displayed intestine-specific expression as previously described (Figs. 7Ba-b and EV7F) (Kim et al, 2004; Romney et al, 2008), and after Δ*cyoB* treatment, a clear increase in *ftn-1p::GFP* expression in the intestine was observed (Figs. 7Bc-d and EV7F). Therefore, these results suggest that this bacteria-induced decrease in labile iron level boosts *ftn-1* expression. Further confirming this idea, Δ*cyoB E. coli*-induced *ftn-1* mRNA (Fig. 7A) and *ftn-1p::GFP* expression in the intestine were both abolished by iron supplementation (Fig. 7Be–h,C). These results together demonstrate that *ftn-1* transcription was increased in response to Δ*cyoB E. coli*-induced labile iron insufficiency. Importantly, we observed that the induction of intestinal *ftn-1* expression by Δ*cyoB E. coli* was partially inhibited by the *bas-1(lf)* mutation (Fig. 7D). Therefore, our findings indicate that dopaminergic and serotonergic neurons promote intestinal *ftn-1* transcription in response to Δ*cyoB E. coli*-induced labile iron reduction.

Furthermore, we investigated the role of this neuron-mediated induction of intestinal ferritin expression in modulating iron metabolism. First, we tested whether this increase in *ftn-1* expression in the intestine was responsible for the reduction in labile iron levels observed in the Δ*cyoB E. coli*-treated worms, as ferritin is well known to sequester labile iron for iron storage. Surprisingly, we found that instead of alleviating labile iron insufficiency, *ftn-1(lf)* caused a further reduction in labile iron levels in the animals treated with Δ*cyoB E. coli*, when compared to the Δ*cyoB E. coli*-treated wild-type worms and the *ftn-1(lf)* mutant worms cultured with wild-type bacteria (Fig. 7E). Therefore, these results suggest an unexpected role of the increased *ftn-1* expression in the Δ*cyoB E. coli*-treated worms in attenuating the reduction in

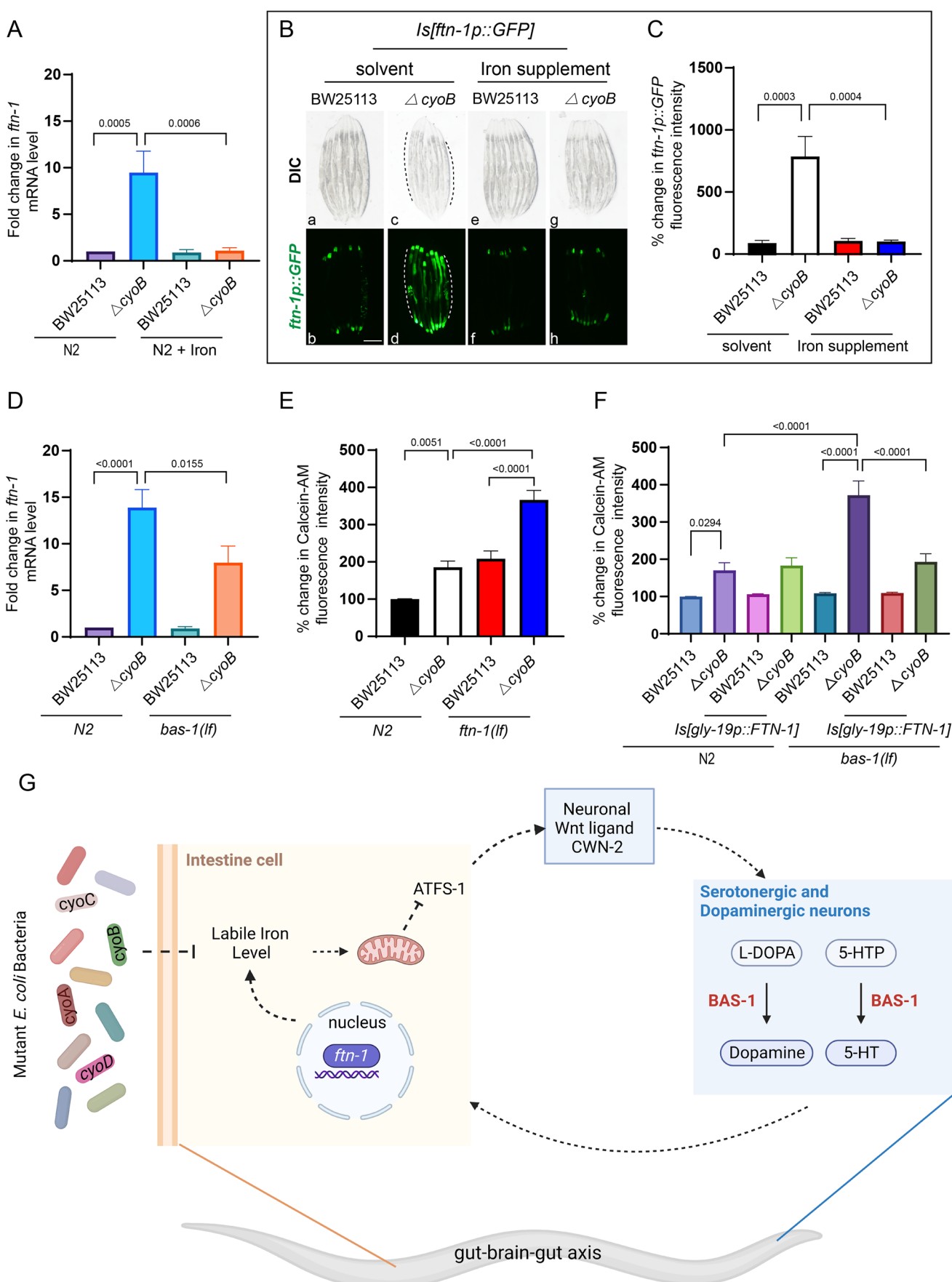

**Figure 7. Increased expression of *bas-1* in dopaminergic and serotonergic neurons promotes intestinal *Ferritin (ftn)-1* expression to attenuate the bacteria-induced reduction in labile iron levels.**

(A) qPCR analyses showing that *ftn-1* transcription is increased in response to Δ*cyoB* bacteria, which is reversed by 4 mM iron supplementation. The *ftn-1* levels in wild-type N2 worms with the indicated treatments were measured. The fold change was calculated by normalization to the *ftn-1* levels in the wild-type N2 worms treated with BW25113 bacteria. $n > 200$ each group. (B, C) Microscopic images and bar graph illustrating that transcription of *ftn-1* is induced in response to the reduction in labile iron caused by Δ*cyoB E. coli*. An integrated transcriptional fusion reporter expressing *ftn-1p::GFP* was used to indicate *ftn-1* transcription. Dashed lines indicate the intestine. FeCl₃ supplementation was performed to recover the labile iron level. Fold change in *ftn-1p::GFP* was calculated by normalizing to the levels in animals treated with both BW25113 and solvent. $n > 20$ for each group. Scale bar, 200 µm. (D) qPCR analyses indicating that the increase in *ftn-1* expression in response to bacteria-induced iron reduction is partially dependent on *bas-1*. The fold change was calculated by normalization to the *ftn-1* levels in the wild-type N2 worms treated with BW25113 bacteria. $n > 200$ each group. (E) Bar graph showing that deletion of *ftn-1* exacerbates the reduction in labile iron levels caused by Δ*cyoB* bacteria. Calcein-AM staining was performed to indicate the labile iron level in the worms with the indicated genotype and treatment. The fold change in Calcein fluorescence intensity was analyzed by normalization to the levels in wild-type N2 animals treated with BW25113 bacteria. $n > 30$ in each group. (F) Bar graph indicating that recovery of *ftn-1* expression in the intestine is able to suppress the *bas-1(lf)*-caused exacerbated reduction in labile iron in the animals treated with Δ*cyoB* bacteria. The *gly-19* promoter was used to drive the intestinal expression of *ftn-1*, and the Calcein fluorescence intensity was quantified to indicate the labile iron level in *C. elegans* with the indicated treatments and genotypes. The fold change in Calcein fluorescence intensity was analyzed by normalization to the levels in wild-type N2 animals treated with BW25113 bacteria. $n > 20$ each group. For all panels, experiments were performed for at least three times. Data are the mean ± SEM. *P* values were determined by one-way ANOVA. (G) Schematic model for a gut–brain–gut axis mediating the role of host brain in surveilling the bacteria that reduce the labile iron level. Source data are available online for this figure.

labile iron levels, which is consistent with the positive effect of *bas-1* in promoting *ftn-1* expression to elevate labile iron levels.

Next, to further test the idea that when confronting Δ*cyoB E. coli*, increased *ftn-1* transcription might actually mediate the role of the dopaminergic and serotonergic neurons in elevating the intestinal labile iron level, we investigated whether overexpressing *ftn-1* in the intestine was sufficient to reverse the *bas-1(lf)*-induced exacerbated reduction in labile iron levels in animals treated with Δ*cyoB E. coli*. We constructed an integration transgene that drives the overexpression of *ftn-1* with an intestine-specific promoter, *gly-19*. Indeed, this intestine-specific expression of *ftn-1* was able to suppress about 50% of the increase of Calcein signal resulting from labile iron insufficiency in *bas-1(lf)* worms cultured with Δ*cyoB E. coli* (Fig. 7F), corroborating that intestinal *ftn-1* acts downstream of the dopamine and serotonin signaling to alleviate bacteria-caused decrease of available iron. Moreover, similar to the effect of *bas-1(lf)*, intestinal *atfs-1* knockdown and neuronal *cwn-2* knockdown also inhibited the induction of *ftn-1* by Δ*cyoB E. coli* treatment (Fig. EV7G,H), further supporting that neuronal signaling triggered by labile iron deficiency and intestinal mitochondrial impairment modulates intestinal *ftn-1* expression. These results together clearly indicate that the increase in *ftn-1* expression in response to Δ*cyoB E. coli* insult acts to promote labile iron level, though we have not been able to illustrate the possible mechanism. In summary, our findings uncover an unexpected gut-brain-gut axis, in which head dopaminergic and serotonergic neurons detect the bacteria-caused insufficiency of labile iron levels through increased *bas-1* expression by perceiving the resultant impairment of the intestinal mitochondria and in turn coordinate the expression of intestinal *ftn-1* to facilitate the elevation of labile iron in the gut (Fig. 7G).

## Discussion

What and how alterations of bacterial activities could be detected by the animal brain and the relevant physiologic significance are underexplored. In this study, by using *C. elegans* as a model system, we reveal that serotonergic and dopaminergic neurons monitor changes in a variety of bacterial processes, providing a comprehensive understanding of what bacterial activity alterations affect the brain. Furthermore, we discovered a gut-brain-gut axis

(intestinal ATFS-1-neuronal BAS-1-intestinal FTN-1) (Fig. 7G), which not only represents a new regulatory mechanism for monitoring and re-establishing iron homeostasis but also exemplifies a bidirectional process underlying the crosstalk between the host nervous system and bacteria.

Our study offers a compelling perspective on how alterations in bacterial activity influence brain function, identifying single deletions in 29 bacterial genes, involved in diverse metabolic processes, that elicit robust responses in serotonergic and dopaminergic neurons of *C. elegans* through a genome-wide screen. Our screen prioritized investigating *E. coli* mutants that induce BAS-1 expression because deficiencies in dopamine and serotonin underlie multiple neurological disorders, and these bacterial strains may offer insights for the development of microbiome-based therapies. According to our knowledge, our study presents the first example of an overall understanding of the impact of bacterial activity on the nervous system and the positive hits from the screen in Table EV1 should provide an important resource for other researchers in the field to identify bacterial factors that modify the dopaminergic and serotonergic neurons. Furthermore, despite intensive research in the field, the detailed cellular and molecular mechanisms underlying bacteria–brain interactions, including the information needed to be communicated, remained largely elusive. Our study identified a gut-brain-gut axis that mediates the bacteria–host communications and also revealed that the level of intracellular labile iron is an important mediator that can convey changes in bacteria to the host nervous system, thus providing mechanistic insights into the microbiota-brain axis. Since the genes in this axis, such as ferritin in iron metabolism, CWN-2 Wnt, and serotonin- and dopamine-related signaling, are highly conserved, it is conceivable that a similar axis may also exist to connect the brain with the gut to modulate iron metabolism as well as to mediate the role of the brain in monitoring bacterial activity changes in other animals.

During the interactions with the host, microbes often compete for the essential iron, a growth-limiting nutrient in the environment (Das et al, 2020; Qi and Han, 2018; Wilson et al, 2016); thus, to counteract the bacteria-induced iron loss, mechanisms have evolved in the host to tightly monitor the iron level. In particular, interorgan regulation of iron metabolism that coordinates iron availability of different tissues is crucial for maintaining systemic

iron homeostasis under this iron scarcity condition, and the liver was the only known organ that is capable of coordinating iron metabolism in different tissues through secreting hepcidin (Michels et al, 2015; Wang and Babitt, 2019). Here, we discovered a surprising role of the nervous system in modulating iron metabolism through a gut-brain-gut axis and that Wnt acts as a previously unknown modulator of systemic iron metabolism, which represents a newly identified interorgan communication mechanism for maintaining iron homeostasis at the whole-animal level. The head neurons monitor iron insufficiency by inspecting the induced mitochondrial impairment, suggesting that this gut-brain-gut axis is likely to detect relatively severe and chronic reductions in iron levels. The gut is the major organ for absorbing iron as well as a battlefield for competing with bacteria for iron; thus, modulating intestinal iron metabolism by the nervous system holds the capacity to maintain body iron homeostasis. To understand the gut-to-brain cell-non-autonomous signaling, we combined transcriptomic screening with functional validation, but have not yet identified any single peptide mediating communication from the gut to the brain (Appendix Fig. S3). This suggests redundancy among peptide families or the presence of alternative mechanisms facilitating gut–brain communication, implicating the complexity of this interaction and the necessity for future detailed investigation.

In addition to insufficiency, excessive body iron also causes severe pathological consequences (Wang and Babitt, 2019), suggesting the existence of neurons for negative regulation of iron levels. Indeed, we found that neurons may use tyramine- and octopamine-mediated signaling to negatively regulate iron levels (Appendix Fig. S2), as eliminating tyramine and octopamine through *tdc-1(lf)* mutation alleviates the labile iron reduction caused by *cyoB* mutant bacteria. The presence of both positive and negative neuronal regulations of iron levels aligns with the evolutionary need for precise iron control. Moreover, our findings not only establish BAS-1-dependent neurotransmitter synthesis as an important mechanism for iron regulation, but also imply the involvement of neuronal electrical activity in modulating peripheral iron metabolism. For example, the *goa-1(sa734)* mutation, which lacks the $G\alpha_o$ subunit and potentiates serotonergic neuronal electrical excitability and serotonin release (Ravi et al, 2021), reverses the $\Delta cyoB$-induced iron decrease (Appendix Fig. S4), as that observed with exogenous serotonin supplementation. This implies that neuronal electrical activity may modulate iron homeostasis by governing neurotransmitter secretion, adding another layer of neuronal regulation of iron metabolism. Therefore, our discovery of the role of the nervous system in monitoring iron metabolism changes in peripheral tissues and mediating the processes of host-bacteria iron communication is highly important, which deepens our understanding of interorgan regulation of systemic iron homeostasis and the mechanisms underlying the surveillance of the gut microbial content by the nervous system.

Surprisingly, we found that ferritin, well-known to sequester and store iron (Harrison and Arosio, 1996), could play an opposite role in elevating free iron levels when worms were treated with the $\Delta cyoB$ E. coli. Actually, increased ferritin expression has also been detected in humans during infection or under iron-insufficient conditions (Birgegard et al, 1978; Weiss et al, 2019), and it is worth exploring whether this increased ferritin level may also act to counter further reduction in iron. These observations implicate that the physiologic context, such as bacterial challenges, may affect

what functions ferritin exerts; however, we have not been able to elucidate the mechanism underlying the role of ferritin in increasing labile iron in this study. Maintaining iron homeostasis is crucial to prevent pathogenesis resulting from iron deficiency or overload; thus, it will be highly important to investigate whether and how bacteria may affect the role of ferritin in modulating labile iron levels, which would deepen our understanding of the mechanism by which ferritin functions.

The impact of gut bacteria on the human brain has been suggested to cause severe pathological consequences, such as neurologic defects. The identification of various bacterial processes affecting head neurons in this study supports the idea that when studying the cause of brain disorders, in addition to analyzing the alterations in the brain, changes in the activities of the associated bacteria and the peripheral tissues may also need to be considered as the original contributors for certain neurologic defects (Morais et al, 2021). Furthermore, dopamine and serotonin are important neurotransmitters whose deficiency results in various neurological defects, such as depression and neurodegeneration, and supplementing the neurotransmitter precursors that cross the blood–brain barrier is clinically used to treat these disorders. The bacterial strains uncovered in this study to stimulate the biosynthesis of serotonin and dopamine may hold great potential for developing a therapeutic strategy for handling neurologic diseases caused by the decrease in serotonin and dopamine. Moreover, our findings may stimulate new studies to investigate how brain defects could cause iron imbalance as well as whether iron insufficiency alters brain activity in other animals. For example, an obvious symptom of iron deficiency anemia in humans is the mysterious pica, which is an unusual eating habit or craving for non-nutritive substances (Coltman, 1969), suggesting that iron insufficiency alters brain activity to modify behavior. In support of this idea, we found that iron deficiency drives worms to seek food with higher iron content (Appendix Fig. S5), underscoring the critical role of dopaminergic and serotonergic neurons in linking iron metabolism with foraging behaviors (D'Agostino et al, 2018; Hills et al, 2004). Our research provides the first mechanistic study showing how iron insufficiency affects head neurons, which may shed light on the connection between iron deficiency and brain disorders.

## Methods

### Reagents and tools table

| Reagent/resource | Reference or source | Identifier or catalog number |
|---|---|---|
| **Experimental models** | | |
| *Is[bas-1p::BAS-1::GFP]* | Shiqing Cai laboratory | *Is[bas-1p::BAS-1::GFP]* |
| Wild-type *C. elegans*, Bristol | CGC | N2 |
| *bas-1(tm351)* | CGC | LC33 |
| *cat-1(e1111)* | CGC | CB1111 |
| *cat-4(e1141)* | CGC | CB1141 |
| *cat-2(e1112)* | CGC | CB1112 |
| *tph-1(n4622)* | CGC | MT14984 |

| Reagent/resource | Reference or source | Identifier or catalog number |
| --- | --- | --- |
| atfs-1(gk3094) | CGC | VC3201 |
| atfs-1(et15) | CGC | QC115 |
| cwn-1(ok546) | CGC | RB763 |
| lin-44(n1792) | CGC | MT5383 |
| sma-6(wk7) | CGC | LT186 |
| ins-7(ot1427) | CGC | OH18835 |
| daf-2(e1370) | CGC | CB1370 |
| tdc-1(n3420) | CGC | MT10661 |
| goa-1(sa734) | CGC | JT734 |
| ftn-1(ok3625) | CGC | RB2603 |
| lin-15B&lin-15A(n765); wuIs177[ftn-1p::GFP + lin-15(+)] | CGC | GA631 |
| hsp-6p::GFP + lin-15(+) | CGC | SJ4100 |
| egl-20(n585) | CGC | MT1215 |
| [rde-1(ne219)]; kbIs7 [nhx-2p::rde-1 + rol-6(su1006)] | CGC | VP303 |
| [sid-1(pk3321)]; pCFJ90 [(myo-2p::mCherry) + unc-119p::sid-1] | CGC | TU3401 |
| cwn-2(ok895) | CGC | VC636 |
| jefEx46[gly-19p::ATFS-1^{Δ1-32 myc}]; Is[bas-1p::BAS-1::GFP] | This study | MAT137 |
| jefIs8[gly-19p::FTN-1::unc-54 3'UTR] | This study | MAT138 |
| jefIs8 [gly-19p::FTN-1::unc-54 3'UTR]; [bas-1(tm351)] | This study | MAT140 |
| [rde-1(ne219)]; [nhx-2p::rde-1 + rol-6(su1006)]; Is[bas-1p::BAS-1::GFP] | This study | MAT141 |
| jefEx46[gly-19p::ATFS-1^{Δ1-32 myc}] | This study | MAT154 |
| [cwn-2(ok895)];Is[bas-1p::BAS-1::GFP] | This study | MAT193 |
| [cat-1(e1111);bas-1(tm351)] | This study | MAT207 |
| [tph-1(n4622);cat-2(e1112)] | This study | MAT208 |
| **Bacterial strains** | | |
| E. coli OP50 | CGC | OP50 |
| E. coli BW25113 | OEC4988 | BW25113 |
| E. coli K-12 Keio Knockout library collection | OEC4988 | K-12 Keio Knockout library collection |
| MRC RNAi library | Source BioScience | N/A |
| ORF RNAi library | GE Dharmacon | N/A |
| **Chemicals and reagents** | | |
| FeCl$_3$ | Sigma-Aldrich | Cat# 157740 |
| FeSO$_4$ | Sigma-Aldrich | Cat# F8633 |
| Dopamine | Sigma-Aldrich | Cat# H8502 |
| Serotonin creatine sulfate | Sigma-Aldrich | Cat# H9523 |
| 2-2' bipyridyl (BP) | Sigma-Aldrich | Cat# D216305 |

| Reagent/resource | Reference or source | Identifier or catalog number |
| --- | --- | --- |
| Calcein-AM | Thermo Fisher | Cat #C3100MP |
| Fluo-4 AM | Beyotime | Cat #S1060 |
| Paraquat | Sigma-Aldrich | Cat #36541 |

## Worm maintenance

All *C. elegans* strains were grown at 20 °C on nematode-growth media (NGM) plates seeded with OP50 unless otherwise indicated. The *daf-2(e1370)* mutant was maintained at 15 °C.

## Screening for the *E. coli* bacterial genes whose single deletion promotes BAS-1::GFP expression in worm neurons and GO analysis

*Bas-1* is specifically expressed in dopaminergic and serotonergic neurons, and the change in BAS-1 expression was analyzed to evaluate the response of these neurons to the presence of bacteria. A BAS-1::GFP translational fusion reporter strain, a gift from Shiqing Cai's lab (Yuan et al, 2020), and the Keio library collection containing 3985 single deletions of all the non-essential genes of *E. coli* were used to screen for the *E. coli* mutants that increase BAS-1::GFP expression of the host. A total of three rounds of screening were performed to determine the bacterial gene deletions that enhance BAS-1::GFP expression. For each round, the wild-type BW25113 *E. coli* and each *E. coli* mutants were cultured in 200 μL liquid broth with 100 μg/mL kanamycin overnight using 96-well plates and then seeded on NGM plates containing 50 μg/mL kanamycin. Subsequently, synchronized L1 worms were cultured, and when they reached the day-2 adult stage, the fluorescence of BAS-1::GFP was acquired using Nikon DS-Qi2 microscopy. ImageJ was used to calculate the BAS-1::GFP fluorescence. Throughout the screen, the BAS-1::GFP worms feeding on wild-type bacteria BW25113 were included for each imaging experiment as a parallel negative control. As previously indicated (Yuan et al, 2020), an increase in BAS-1::GFP by ≥10% relative to animals feeding with parental BW25113 wild-type bacteria was considered to be significant. The 432 bacterial clones identified as increasing BAS-1::GFP expression in the first round were re-examined in the second screening round. Then, the 70 positive clones from the second round were included in the third round, and 29 positive clones were observed to enhance BAS-1::GFP expression in all three independent experiments. To explore what bacterial processes would affect host BAS-1::GFP expression, GO analyses for these 29 mutant genes were then performed by clusterProfiler, and corresponding images were generated by ggplot2 (v3.3.0).

## Imaging and analysis of the expression of transgenic reporters

For microscopic analyses, approximately 150 L1-staged worms expressing indicated transgenic reporters were cultured with indicated *E. coli* bacteria, and when the worms reached the day-2 adult stage, worms were randomly picked and anesthetized with

150 mg/mL levamisole on a 2% agarose pad before imaging. For sample randomization, the worms analyzed by complex microscopy were randomly selected under a dissection microscope. Fluorescence levels in the head neurons of BAS-1::GFP-expressing worms in all the experiments, except the ones from the screen in Figs. 1B,C and EV1 that were scored by using Nikon DS-Qi2 microscopy, were acquired by confocal microscopy (Zeiss LSM 880 NLO, 40×), and ImageJ software was used to quantify the fluorescence intensity. To analyze the expression of *ftn-1p::GFP* and *hsp-6p::GFP* resulting from specific bacterial treatments, pick and anesthetize worms on NGM plate with 8–10 animals per group to image with a Nikon SMZ18 or mount worms on slides for imaging using a Nikon DS-Qi2 microscope. The intensity of GFP fluorescence signal was quantified with ImageJ.

## Analyzing levels of dopamine and serotonin by HPLC-MS

The dopamine and serotonin levels in *C. elegans* were detected by using HPLC-MS. Approximately 2000–10,000 day-2 adult worms treated with BW25113 or $\Delta cyoB$ *E. coli* were collected and washed with M9 buffer. Add 80% HPLC-grade methanol (pre-chilled at $-80\,^{\circ}C$) to the pelleted worms with the volume/weight ratio at $10\,\mu L/mg$ and then homogenize the worms with FastPrep-24 (MP Biomedicals) for 30 s. Next, transfer the worm lysates to a new Eppendorf tube and incubate at $-80\,^{\circ}C$ overnight to allow extract the dopamine and serotonin. Then, centrifuge these samples at $14{,}000\times g$ for 20 min at $4\,^{\circ}C$ and transfer the supernatant that contains the dopamine and serotonin components to a new Eppendorf tube. In total, $10\,\mu L$ of these supernatants were used to determine the protein level using the BCA Protein Assay Kit (E162-01). The rest of the supernatants were lyophilized in a vacuum centrifugal concentrator (MS-SP102), and the dried powder was resuspended in $60\,\mu L$ of 80% methanol followed by vigorous vortexing. After centrifugation at $14{,}000\times g$, $10\,\mu L$ of the resulting supernatant was analyzed by HPLC–MS to determine the dopamine and serotonin levels. A standard curve of the commercially obtained standards (Sigma, USA) was used to determine the dopamine and serotonin concentrations.

## Chemical supplementation

$FeCl_3$, $FeSO_4$, dopamine, and serotonin creatine sulfate dissolved in water, and 2-2' bipyridyl (BP) dissolved in DMSO were added to the NGM before pouring to final concentrations of 4 mM, 4 mM, 12.5 mg/mL, 10 mg/mL, and 50 μM, respectively. An equivalent volume of the corresponding solvent was added to the NGM as a negative control. To assay the impact of these chemicals on the indicated phenotypes, L1-staged worms were cultured on NGM plates with the indicated chemical supplements and bacterial treatments, and when they reached the day-2 adult stage, the indicated phenotypes were scored as described.

## Labile iron detection by Calcein-AM

The level of intracellular labile iron in *C. elegans* was analyzed with the widely used Calcein-AM dye as previously described (Qi and Han, 2018). Approximately 20–30 day-2 adult worms with the indicated genotype and bacterial treatment were randomly picked under a dissection microscope. After removing the residual

bacteria, these worms were then transferred to $100\,\mu L$ of the commonly used M9 buffer (3.0 g $KH_2PO_4$, 6.0 g $Na_2HPO_4$, 0.5 g NaCl, 1.0 g $NH_4Cl$ dissolved in 1 L $H_2O$) containing $0.5\,\mu g/mL$ Calcein-AM and incubated at $20\,^{\circ}C$ for labile iron staining. Next, after rinsing with M9 buffer to remove the residual Calcein-AM dye, these worms stained with Calcein-AM were mounted on an agarose pad (2% agarose). To exclude the effects of cell permeability on Calcein-AM staining, Fluo-4 AM was used as a control. *C. elegans* were stained with 0.5 μg/mL Fluo-4 AM, following the same procedure as for Calcein-AM staining. The Calcein-AM and Fluo-4 AM fluorescence signals were detected using a fluorescence microscope (Nikon DS-Qi2, 40X) with DIC and the 488 nm excitation channel. Calcein and Fluo-4 AM fluorescence were quantified using ImageJ by measuring mean intensity in the images. To ensure consistency and comparable quantification across different groups, the same threshold was used for all microscopic images during analyses.

## Calcein-AM fluorescence image data analysis and normalization procedure

The analysis of Calcein-AM fluorescence images was performed using ImageJ software. First, the FITC channel images were converted to 8-bit format. A consistent threshold was then applied uniformly across all images within the same biological replicate to detect and quantify Calcein-AM fluorescent signals. Next, fluorescence intensity within the defined regions of interest was measured using ImageJ's built-in analysis tools. To assess changes in fluorescence intensity in the treatment group relative to the control, data were normalized as follows: the mean fluorescence intensity of the control group was calculated, and each sample's raw fluorescence value was divided by this mean to obtain a normalized ratio relative to the control. Finally, these ratios were multiplied by 100 to express the results as percentages, which were subsequently used for statistical analysis.

## Measurement of basal oxygen consumption rate (OCR)

The basal OCR of worms was analyzed by a Seahorse XFe96 analyzer with an extracellular flux assay kit as previously described (Koopman et al, 2016). Specifically, 200 adult worms with the indicated genotypes and treatments were collected for each group after the residual bacteria were removed by washing with M9 buffer twice. Then aliquot 10–30 worms/well to a Seahorse XF96 cell culture microplate. Approximately 5–8 technical replicates per independent experiment and at least three biological replicates were performed. The basal OCR in live *C. elegans* was probed using the following program: five cycles of mix for 2 min, wait for 0.5 min, measure for 2 min. The seahorse measurements of OCR were normalized by worm numbers.

## RNA interference in *C. elegans*

RNAi experiments were carried out by feeding the worms with bacteria that generate dsRNA against corresponding worm genes as described previously (Tang and Han, 2017). Specifically, the indicated RNAi HT115 *E. coli* bacteria from the MRC RNAi library and ORF RNAi library were cultured in LB medium containing 100 μg/mL ampicillin and 15 μg/mL tetracycline for

8–10 h at 37 °C, and then 350 μL RNAi bacteria were seeded onto NGM plates with 2 mM IPTG. Then, these plates with the indicated RNAi bacteria were maintained at room temperature for 3–5 days to allow the production and accumulation of dsRNA before the worms were seeded. Approximately 100–200 L1-staged worms with the indicated genotypes and treatments were cultured on these plates, and the day-2 adults were scored. HT115 bacteria containing the empty PL4440 vector were used as the RNAi negative control. The VP303 worm strain, which expressed *rde-1* only in the intestine (Espelt et al, 2005), was used to perform intestine-specific RNAi analyses. While the TU3401 *C. elegans* strain, which overexpressed *sid-1* by neuron-specific promoter in the *sid-1(lf)* mutant, displaying neuron-specific RNAi efficacy (Calixto et al, 2010), was used in experiments that aimed to knock down genes specifically in the neurons.

## Analyses of worm survival rate after paraquat treatment

The rate of worms surviving paraquat treatment was analyzed to evaluate the animals' ability to cope with mitochondrial stress as previously described (Schaar et al, 2015). Briefly, L1-staged worms with the indicated genotypes were grown on NGM plates containing BW25113 *or Δ cyoB E. coli* with and without FeCl$_3$ supplementation (4 mM) at 20 °C. When they reached the late L4 stage, these worms were transferred to NGM plates supplemented with 5 mM paraquat containing OP50 bacteria. The survival of animals on paraquat plates was scored daily until all worms were dead, which was evaluated by the animals showing no response to a light touch with a platinum wire. If worms died of unnatural causes, such as desiccation on the side of the dish, they were excluded from the scoring. The mean survival time and the statistics were analyzed by GraphPad Prism. Mean survival time was calculated as the sum of the survival time of each worm with the indicated genotype and treatment/total number of worms of the corresponding genotype and treatment.

## Identification of FTN-1 and FTN-2 protein levels by LC-MS/MS

Approximately 10,000 synchronized worms with indicated treatments were collected, washed with M9 buffer, and flash-frozen in liquid nitrogen. The worms were lysed using RIPA lysis buffer, followed by sonication. The extracted protein was quantified using BCA assays (Catalog No. 23225, Thermo Scientific). Twenty-five micrograms of total protein were purified using SP3 technology (Hughes et al, 2019) and digested with trypsin (Catalog No. V5280, Promega) at a 1:100 ratio (w/w, enzyme/protein) at 37 °C overnight. The digested products were lyophilized and prepared for LC-MS/MS analysis. The lyophilized peptides were resuspended in 30 μL of 0.1% formic acid, and 1 μL aliquots were injected using the nanoElute UHPLC System (Bruker). The eluted peptides were analyzed with a TIMS quadrupole time-of-flight timsTOF Pro 2 instrument (Bruker Daltonics) using a CaptiveSpray nano-electrospray source (Hughes et al, 2019). The dda-PASEF mode was employed with a *m/z* range of 300–1500 and a 1/K0 range of 0.75–1.3, with a Ramp Time of 166 ms. LC-MS/MS data files were processed using the Parallel Search Engine in Real-time (PaSER version 2023) against the *C. elegans* proteome from Uniprot (https://www.uniprot.org/). Search parameters included a peptide

mass tolerance of 20 ppm for 1 isotopic peak, a precursor mass range of 600–6000 Da, and tryptic digestion specificity. Variable modifications included oxidation of methionine and carbamido-methylation of cysteine. Normalization was enabled for label-free quantification, and the False Discovery Rate (FDR) was set to 1% for both protein and peptide levels.

## Generation of transgenic strains

The *Is[gly-19p::FTN-1::unc-54 3'UTR]* and *Ex[gly-19p::ATFS-1$^{Δ1-32\ myc}$]* transgenes specifically expressing *ftn-1 and atfs-1(gf)* in the intestine were used to evaluate the role of ferritin in alleviating free iron reduction and the impact of the mitochondrial state on modulating *bas-1* expression and free iron levels, respectively. The promoter of *gly-19* was used to drive the expression of these genes specifically in the intestine (Shao et al, 2016). The 5'-TTTTTTTCAGTACATTTTTTCATTTCGGGTTCA-3' and 5'-CTGGAAATTTAAATTTAATTCTTTGGAATTTAGT-3' primers were used to generate the promoter of *gly-19* by PCR, and *ftn-1* cDNA was amplified by using the 5'-ATGTCTCTAGCTCGTCAAAACTATC-3' and 5'-TTAATCAGAAAATTCCTCTTTGTCGAAC-3' primers. The plasmid used for intestine-expression of *atfs-1(gf)* was obtained from Ye Tian's lab. A DNA mixture containing 20 ng/μL *myo-2p::mCherry* co-injection marker and 50 ng/μL plasmid with intestine-specific expression of the indicated genes was microinjected to obtain the extrachromosomal transgenes. The worms carrying *gly-19p::FTN-1::unc-54 3'UTR* extrachromosomal array were used for the construction of *Is[gly-19p::FTN-1::unc-54 3'UTR]* integration transgenes by using the UV irradiation method.

## Quantitative RT-PCR (qPCR) analyses

To perform qPCR measurements of gene transcription, more than 150 young adult worms with the indicated genotypes and treatments were collected for RNA extraction by using a MicroElute Total RNA kit. Approximately 300–800 ng extracted RNA was treated with One-Step gDNA Remover (Transgene) to remove potential genomic DNA contamination and then used to produce complementary DNA by reverse transcription with the cDNA Synthesis SuperMix kit (Transgene). The generated cDNA was used as the template for qPCR analysis by utilizing Universal qPCR Master Mix (BioLabs). At least three independent experiments were performed to indicate the change in gene transcription. The transcription of a housekeeping gene, *rpl-26*, was used as an internal control as previously described (Tang and Han, 2017). The primers for real-time PCR were as follows:

| Gene name | Primers | Sequence |
| --- | --- | --- |
| *rpl-26* | Primer F | CTTCGAAGGTCGCTATCACCA |
| | Primer R | TTGACGGTCTCGTCAGTGTG |
| *bas-1* | Primer F | TTGGTTGCACGAGCTTCAGT |
| | Primer R | GTGAGCCTGGTCAGAGCAAT |
| *ftn-1* | Primer F | CTATCACGATGAAGTCGAAGC |
| | Primer R | GGCAATGTTCCGAAGTGCG |
| *smf-1* | Primer F | TCGGAATGTTGCTCCAACGA |

| Gene name | Primers | Sequence |
|-----------|---------|----------|
|  | Primer R | GCCAGAGAATGATGCGAGGA |
| *smf-2* | Primer F | TGAGCTCTGAAAATGCCTGGT |
|  | Primer R | AAGCTTACGCCAGGAGAACC |
| *smf-3* | Primer F | GGAGTGCGAAAGTTTGAAGC |
|  | Primer R | TTGACAAGTGCCGAGTGAAG |
| *cwn-2* | Primer F | TCGACATTCTGGAACGTTAGGA |
|  | Primer R | GCACAAGCTATCACAGCCTTC |

## Statistical analysis

To avoid potential bias, the information on treatment and genotypes was masked before the phenotypes were scored. At least three independent experiments were performed for all the results. The sample size ("*n*" number) indicated in the Figures and "Results" is the number of worms scored in that experiment. All statistics were analyzed by using GraphPad Prism. Statistical comparisons between two groups were analyzed by Student's *t* test. Comparisons among multiple groups were analyzed by one-way ANOVA or two-way ANOVA, respectively. Data are presented as the standard error of the mean.

## Data availability

This study includes no data deposited in external repositories.

The source data of this paper are collected in the following database record: biostudies:S-SCDT-10_1038-S44318-025-00619-6.

## Peer review information

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

## Acknowledgements

We thank the CGC (funded by NIH [P40OD010440]) for *C. elegans* strains, S Cai (Institute of Neuroscience, Chinese Academy of Sciences), Y Tian (Institute of Genetics and Developmental Biology, Chinese Academy of Sciences) and Y Liu (Peking University) for providing *C. elegans* strains and reagents, and P Zhao (Westlake University), S Wu (Westlake University), F Jin (Westlake University), the High-performance Computing Center of Westlake University and the Microscopic Imaging Center of Westlake University for technical support. We thank X Xu and M Fan from the Mass Spectrometry & Metabolomics Core Facility of Westlake University for HPLC-MS analysis. This research was supported by the National Natural Science Foundation of China (No. 32350015, No. 32530030, No. 32070565, and No. 31871465), the National Key Research and Development Program of China (No. 2019YFA0802900), the HRHI program (No. 202209003 and No. 202109007) of the Westlake Laboratory of Life Sciences and Biomedicine, the Zhejiang Provincial Natural Science Foundation of China (LQN25C070002 and No. LQ23C040002), the Westlake Education Foundation of Westlake University and the Key Laboratory of Growth Regulation and Translational Research of Zhejiang Province. Hongyun Tang is the senior author.

## Author contributions

**Guanqun Li**: Conceptualization; Data curation; Formal analysis; Funding acquisition; Investigation; Visualization; Methodology; Writing—original draft; Project administration; Writing—review and editing. **Yangyang Wu**: Conceptualization; Data curation; Formal analysis; Validation; Investigation;

Methodology; Writing—original draft; Project administration; Writing—review and editing. **Xiaowen Huang**: Data curation; Formal analysis; Investigation; Methodology; Writing—original draft; Writing—review and editing. **Minghui Du**: Investigation; Methodology. **Hongyun Tang**: Conceptualization; Resources; Software; Supervision; Validation; Visualization; Methodology; Writing—original draft; Project administration; Writing—review and editing.

Source data underlying figure panels in this paper may have individual authorship assigned. Where available, figure panel/source data authorship is listed in the following database record: biostudies:S-SCDT-10_1038-S44318-025-00619-6.

## Disclosure and competing interests statement

The authors declare no competing interests.

# Expanded View Figures

**Figure EV1.** **Effects and categories of the bacterial genes that promote BAS-1 expression in *C. elegans* dopaminergic and serotonergic neurons following single deletion.** ▶

(**A**) Graph showing the increase of BAS-1::GFP induced by the 29 *E. coli* mutants positive hits obtained from the screen described in Fig. 1B. % change in BAS-1::GFP fluorescence intensity were normalized to the expression levels in worms treated with wild-type BW25113 *E. coli*. Each dot indicates the effect of one mutant *E. coli* on the BAS-1::GFP level. $n >= 18$ each group. *P* values were determined by unpaired two-tailed *t* test. (**B–D**) Gene ontology analysis showing the enrichment of the screen hits in a variety of bacterial processes, including transportation and metabolism. GO analyses indicate the categories of the positive hits (**B**). Further GO analysis shows the detailed metabolic processes (**C**) and transportation processes (**D**) in which the bacterial genes, whose inactivation promotes BAS-1::GFP expression, are involved. (**E**) Bar graphs of qPCR analyses showing that *bas-1* mRNA levels are increased in the *Is[bas-1p::BAS-1::GFP]* worms treated with Δ*cyoB E. coli*. *Is[bas-1p::BAS-1::GFP]* animals treated with BW25113 were used as the control to calculate the fold change in mRNA levels. Data are the mean ± SEM. *P* values were determined by unpaired *t* test. $n > 200$ each group. (**F–K**) Representative chromatograms illustrate the acquisition time, ion pairs, and ion counts for serotonin (**F–H**) and dopamine (**I–K**) detection. (**F–H**) Present chromatograms of the serotonin commercial standard, serotonin in *C. elegans* fed BW25113, and serotonin in Δ*cyoB*-fed worms, respectively. (**I–K**) Depict the dopamine commercial standard, dopamine in BW25113-fed worms, and dopamine in Δ*cyoB*-fed worms, respectively. All panels exhibit retention time on the *x* axis versus ion counts on the *y* axis, with insets displaying the parent→product ion transitions (*m/z*) utilized for quantification.

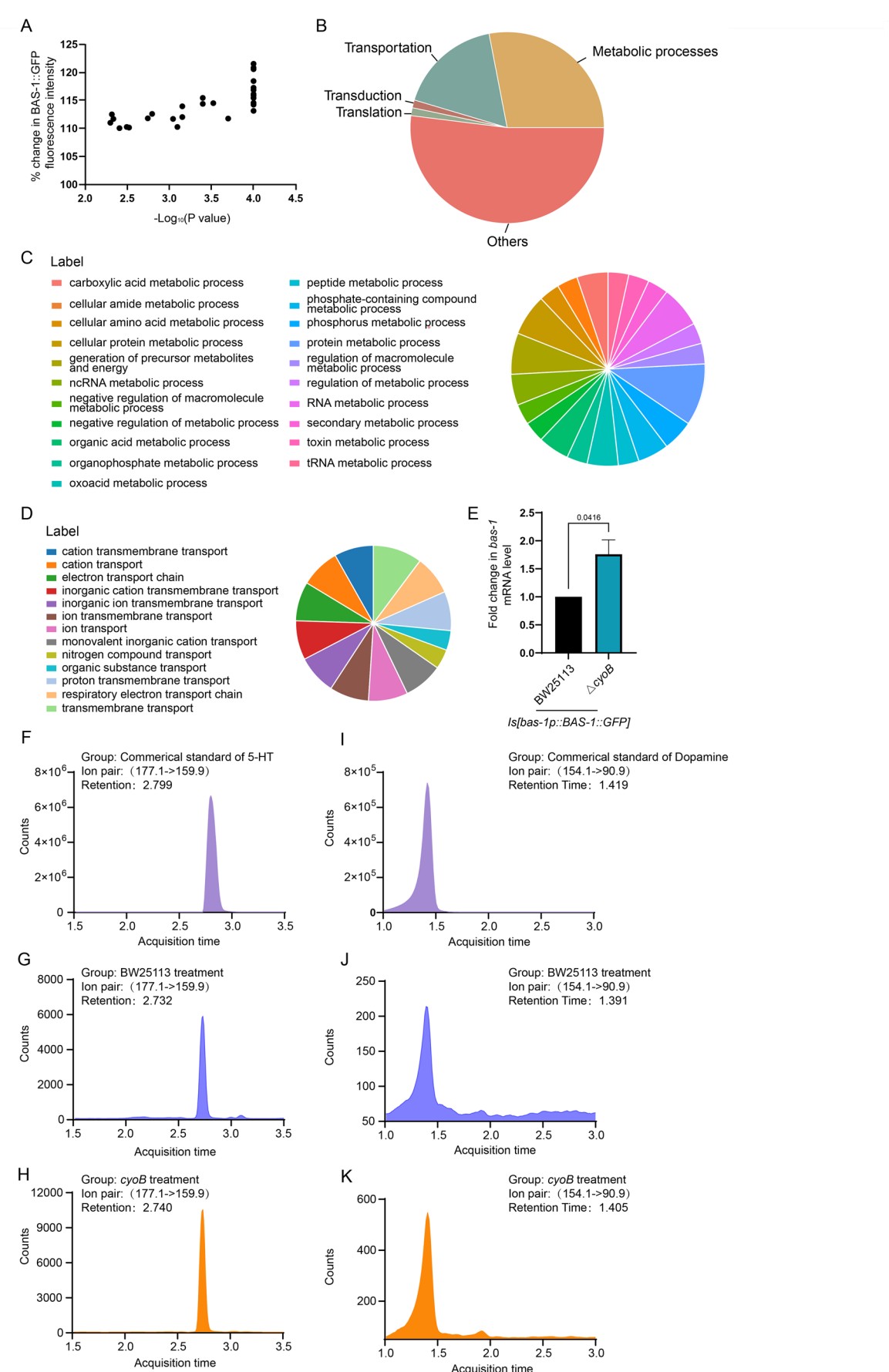

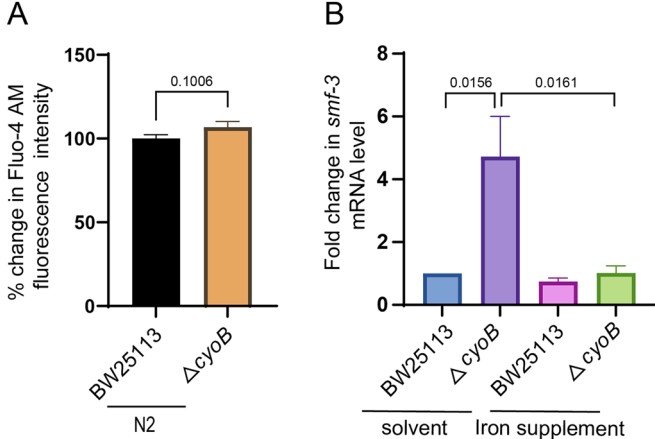

**Figure EV2.   The Δ*cyoB E. coli* treatment induces *smf-3* transcription, which is suppressed by iron supplementation.**

(A) Bar graphs indicating no changes in Fluo-4 AM fluorescence in the host *C. elegans* treated with Δ*cyoB E. coli* compared to those treated with BW25113 *E. coli*. Data represent the mean ± SEM. ns > 0.05 by unpaired *t* test. *n* > 30 in each group. (B) qPCR results showing that transcription of *smf-3*, encoding transporters for iron absorbing in *C. elegans* (Romney et al, 2011), is increased in response to Δ*cyoB E. coli* treatment. 4 mM FeCl$_3$ was used for iron supplementation. Wild-type animals treated with BW25113 *E. coli* were used as the control to calculate the fold change in *smf-3* mRNA levels. Data are the mean ± SEM. *P* values were determined by one-way ANOVA; *n* > 200 each group.

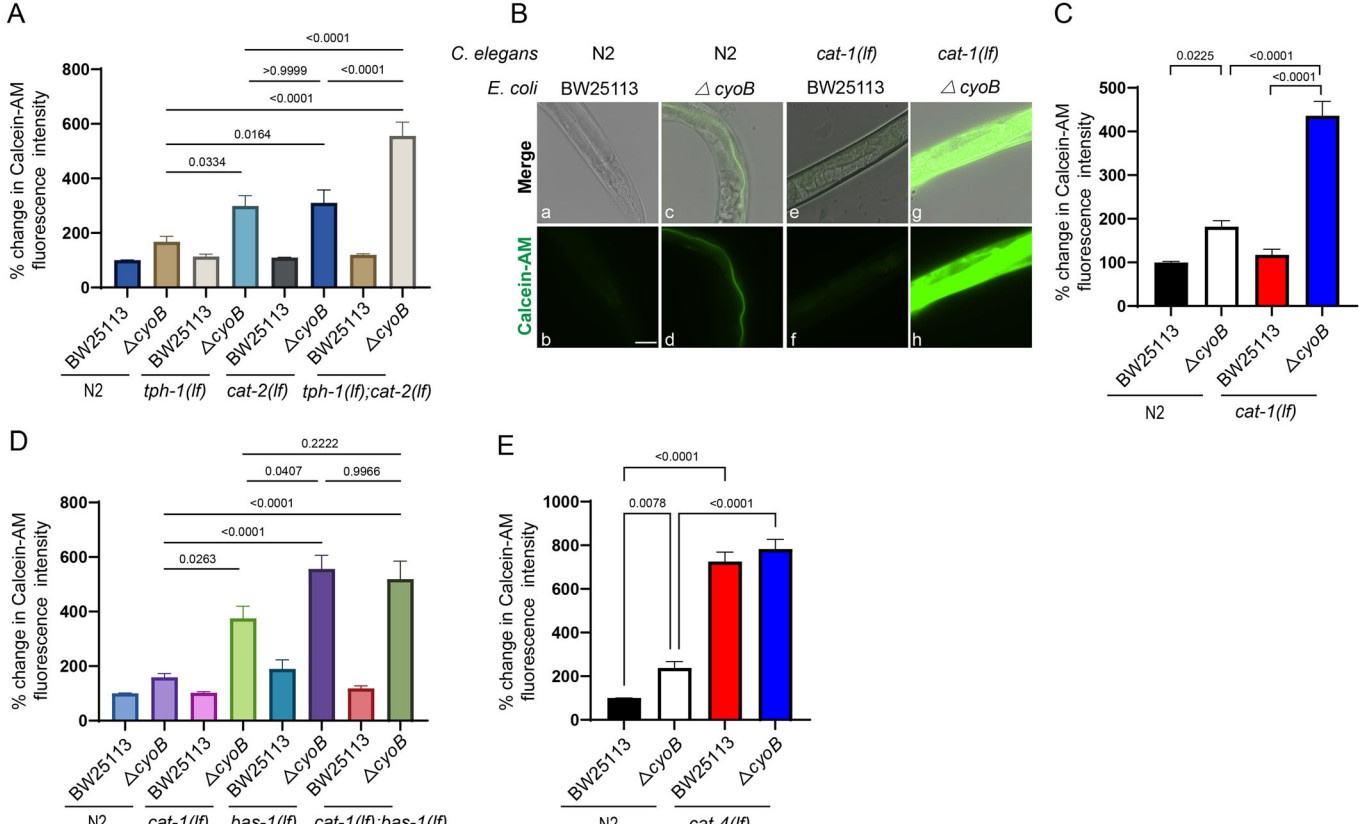

**Figure EV3.  Inhibiting dopamine and serotonin release by blocking their import into vesicles exacerbates the reduction in labile iron caused by ΔcyoB E. coli.**

(**A**) Bar graphs indicating that both *tph-1(lf)* and *cat-2(lf)* mutations exacerbated iron reduction induced by ΔcyoB *E. coli*. *tph-1(lf);cat-2(lf)* displays a stronger reduction in labile iron compared to each single mutant. The *Y* axis represents the percentage change in Calcein fluorescence, normalized to the levels observed in wild-type N2 animals treated with BW25113 bacteria. (**B, C**) Images and bar graph indicating that the *cat-1(lf)* mutation caused a further reduction in the labile iron level in the presence of ΔcyoB. The % change in the Calcein fluorescence was calculated by normalization to the levels in wild-type N2 animals treated with BW25113 bacteria. CAT-1 is required for transporting monoamines, including dopamine and serotonin, into vesicles for release. Scale bar, 50 µm. (**D, E**) Bar graphs depicting the Calcein signal in worms with indicated genotypes and bacterial treatments. The percentage change in Calcein fluorescence was calculated by normalizing to the levels observed in wild-type N2 animals treated with BW25113 bacteria. For all panels, experiments were performed for at least three times. Data are the mean ± SEM. *P* values were determined by one-way ANOVA; *n* > 25 each group.

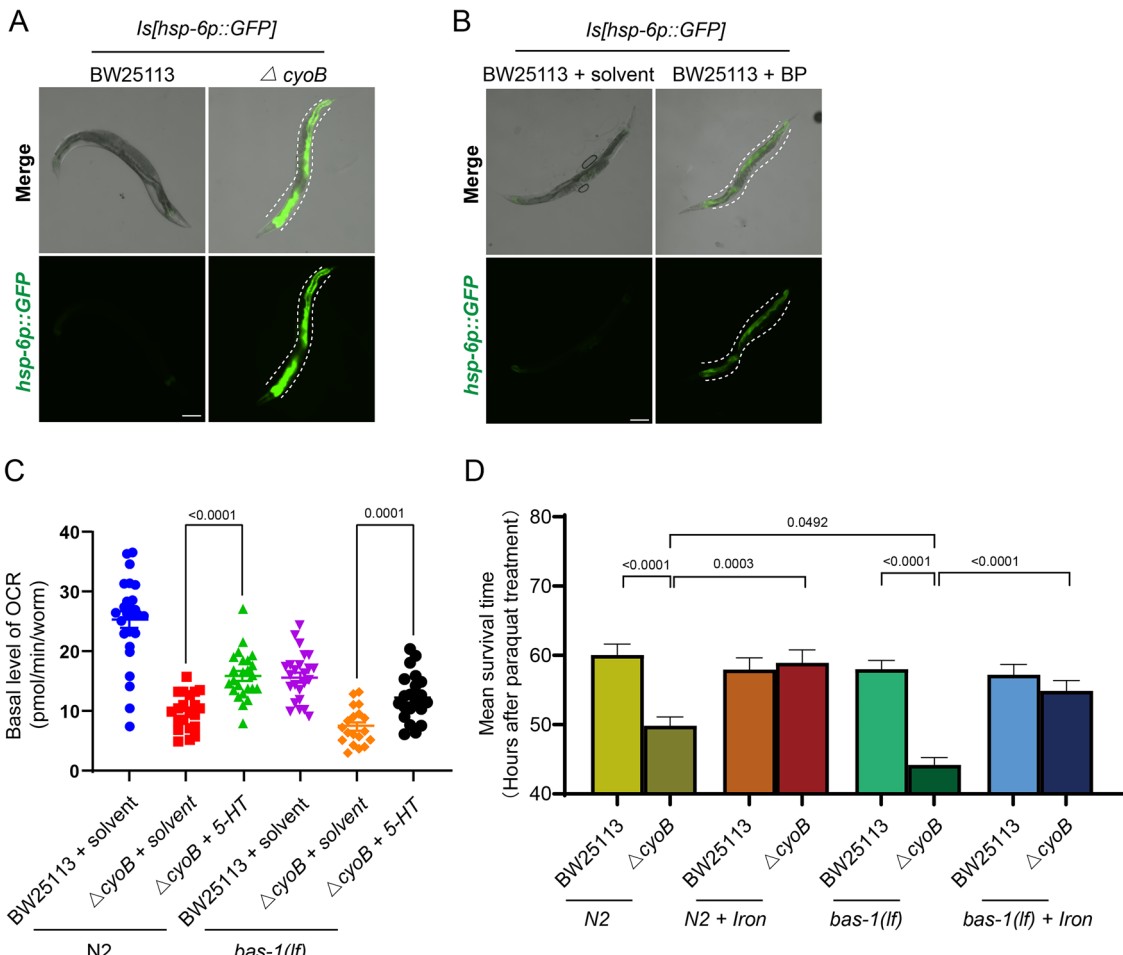

**Figure EV4. The intestinal mitochondria impairment induced by the Δ*cyoB E. coli* and iron chelator treatment and the role of serotonergic and dopaminergic neurons response to the presence of Δ*cyoB E. coli* in preventing decline in mitochondrial function.**

(A) Microscopic images indicating that Δ*cyoB* mutant *E. coli* mainly induces mitochondrial stress in the intestine of *C. elegans*. The induction of *hsp-6p::gfp* expression in the intestine (white lines) was detected in the presence of Δ*cyoB E. coli*. Scale bar, 100 μm. (B) Images showing that the iron chelator BP (25 μM) induces *hsp-6p::gfp* expression mainly in the intestine of *C. elegans*. White lines indicate the location of the intestine. Scale bar, 100 μm. (C) Bar graphs depicting the reduction in basal oxygen consumption rate (OCR) in wild-type N2 and *bas-1(lf)* worms treated with Δ*cyoB E. coli*, which was partially reversed by serotonin (5-HT) supplementation. Each dot represents the OCR normalized to each worm. P values were determined by unpaired *t* test; *n* > 50 per group. (D) Bar graph showing that the increase in *bas-1* expression in response to Δ*cyoB* bacteria is important for preventing a further decline in mitochondrial function. Paraquat was used to challenge mitochondrial function, and the mean survival time of *C. elegans* with the indicated genotypes and treatments was calculated as described in the Methods. Data are the mean ± SEM. *P* values were determined by one-way ANOVA; *n* > 120 each group.

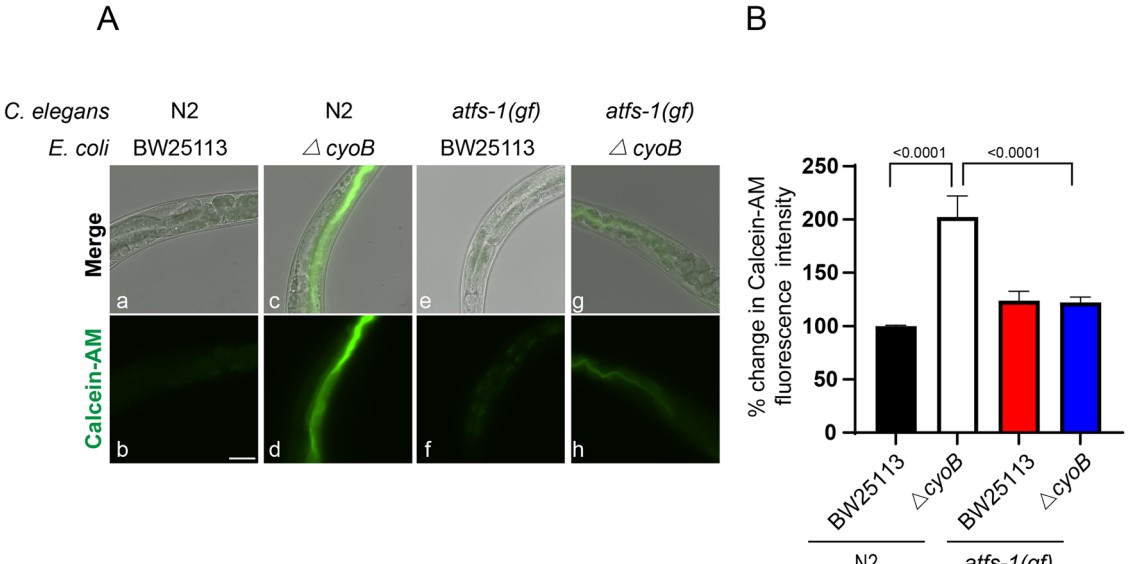

**Figure EV5. Over-activation of ATFS-1 is able to alleviate the reduction in labile iron levels caused by ΔcyoB E. coli.**

(A, B) Images and bar graph indicating that the dominant active form of ATFS-1 suppressed the reduction in labile iron level caused by the ΔcyoB E. coli treatment. Calcein-AM staining was performed in N2 and atfs-1(gf) worms with the indicated bacterial treatments to indicate the labile iron levels. The % change in Calcein fluorescence was calculated by normalization to the levels in N2 worms treated with BW25113. Data are the mean ± SEM. P values were determined by one-way ANOVA. n > 25 each group. Scale bar, 50 μm.

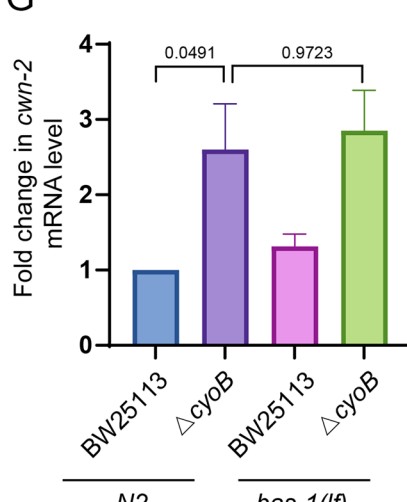

◀ **Figure EV6. Wnt ligands are required for alleviating iron deficiency caused by the Δ*cyoB E. coli* mutant.**

(**A–F**) Bar graphs depicting the change in Calcein signal in worms with the indicated genotypes and treatments. The fold change was calculated by normalizing to the levels in N2 worms treated with BW25113. *P* values were determined by one-way ANOVA. *n* > 30 per group. (**G**) qPCR analyses indicate that deletion of *bas-1* shows no effect on the transcription of *cwn-2*, supporting that *bas-1* functions downstream of *cwn-2*. The mRNA of *cwn-2* in worms with the indicated genotypes and treatments was analyzed by qPCR. In the presence of Δ*cyoB E. coli* mutant bacteria, the *bas-1(lf)* worms displayed similar levels of *cwn-2* mRNA to that in wild-type N2 *C. elegans*. N2 wild-type animals treated with BW25113 was used as the control to calculate the fold change in mRNA levels. *P* values were determined by one-way ANOVA. *n* > 200 each group.

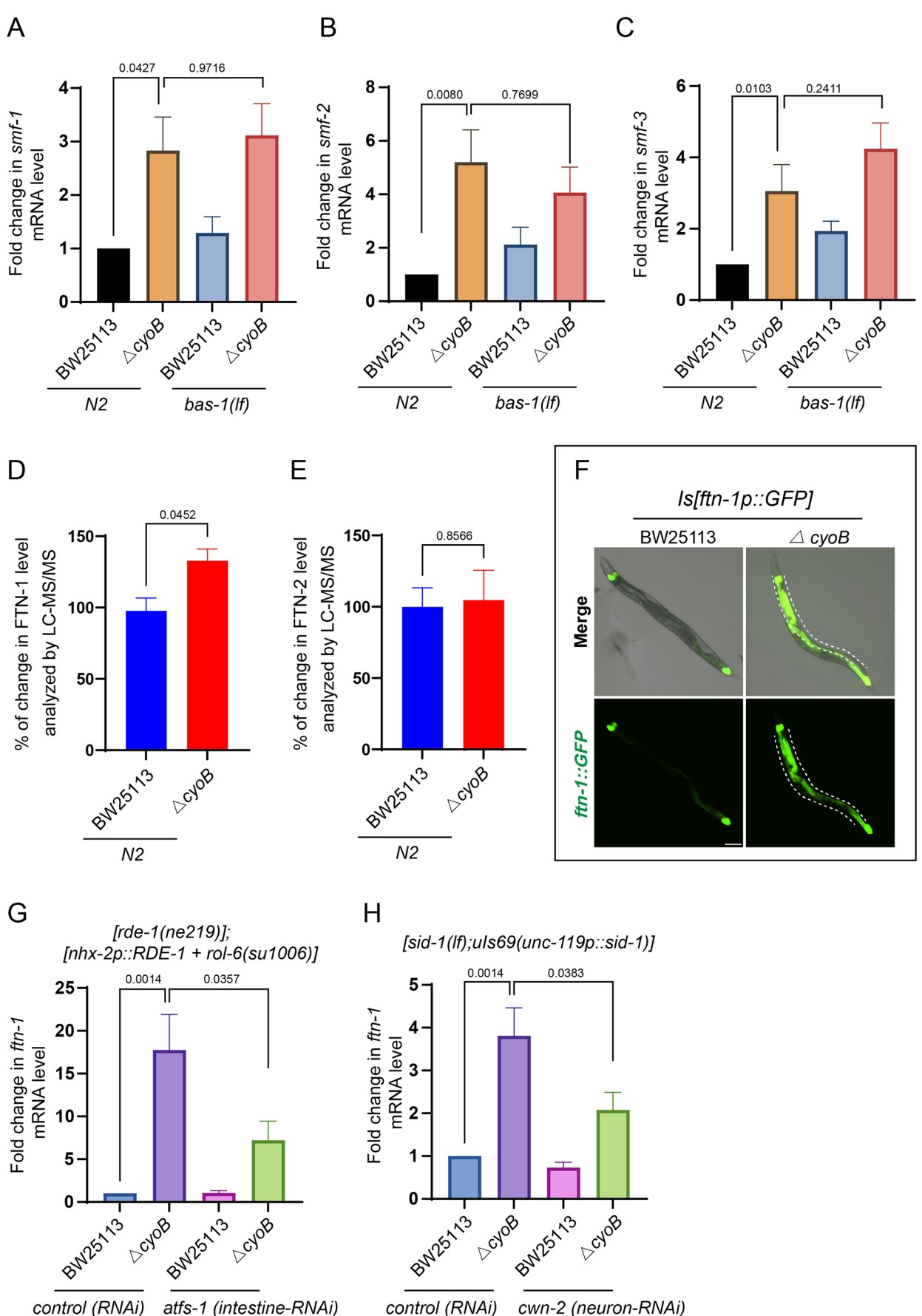

◀ **Figure EV7. In response to the Δ*cyoB E. coli* mutant, increased transcription of ferritin but not iron absorption genes is dependent on the host neuronal response.**

(A–C) qPCR results show that *bas-1* is not required for the induction of *smf-1/-2/-3* mRNA expression caused by the Δ*cyoB E. coli*. The transcription of *smf-1/-2/-3*, genes encoding transporters for absorbing iron in *C. elegans* (Romney et al, 2011), are increased in response to Δ*cyoB E. coli*. In the presence of Δ*cyoB E. coli*, the *bas-1(lf)* worms showed no significant difference in the level of *smf-1/-2/-3* mRNA compared with N2 wild-type *C. elegans*. Wild-type animals treated with BW25113 was used as the control to calculate the fold change in mRNA levels. $n > 200$ each group. (D, E) Bar graph showing LC-MS/MS analyses of FTN-1 and FTN-2 protein levels in *C. elegans* treated with BW25113 or Δ*cyoB E. coli*. The percent change of FTN-1 (D) or FTN-2 (E) was calculated by normalizing to the levels from the *C. elegans* treated with BW25113. $n > 500$ each group. *P* values were determined by unpaired t test. (F) Microscopic images showing the intestine-specific expression of *ftn-1* and its induction by the presence of Δ*cyoB*. The induction of the Is[*ftn-1p::GFP*] was observed with Δ*cyoB* treatment when compared to the BW25113 group. The intestine was outlined. Scale bar, 100 μm. (G) qPCR results show that intestine *atfs-1* is at least partially required for the increase in *ftn-1* transcription in response to Δ*cyoB E. coli* mutant bacteria. The VP303 worm strain was used to perform intestine-specific knockdown of *atfs-1* in the presence of the indicated bacteria. Fold change was determined by normalization to the animals fed with both control RNAi and BW25113 bacteria. $n > 200$ each group. (H) qPCR analyses show that neuronal *cwn-2* is at least partially required for the induction of *ftn-1* mRNA expression in the presence of Δ*cyoB E. coli* mutant bacteria. TU3401[*sid-1(lf); uIs69(unc-119p::sid-1)*] was used to knock down *cwn-2* specifically in neurons. Δ*cyoB E. coli*-induced *ftn-1* transcription is partially suppressed by *cwn-2(neuron-RNAi)*. $n > 200$ each group. For (A–C, G, H), data are the mean ± SEM. *P* values were determined by one-way ANOVA.

