## [Peer Review File · The EMBO Journal]

A gut-brain-gut axis orchestrates host responses counteracting microbiome-induced iron insufficiency

Guanqun Li, Yangyang Wu, Xiaowen Huang, Minghui Du, and Hongyun Tang

Corresponding author: Hongyun Tang (tanghongyun@westlake.edu.cn)

Review Timeline:

Submission Date:	3rd Dec 24
Editorial Decision:	9th Dec 24
Appeal:	12th Dec 24
Editorial Decision:	20th Feb 25
Revision Received:	20th May 25
Editorial Decision:	25th Sep 25
Revision Received:	30th Sep 25
Accepted:	10th Oct 25

Editor: Ieva Gailite

Transaction Report:

Dear Dr. Tang,

Thank you for submitting your manuscript "A gut-brain-gut axis orchestrates host response to counteract bacteria-caused iron insufficiency" to The EMBO Journal. I have now read your manuscript in detail and discussed the work with other members of the editorial team. I regret to inform you that we have decided not to pursue the publication at The EMBO Journal.

We appreciate that your study describes activation of dopamine and serotonin production in the head neurons of *C. elegans* exposed to *E. coli* carrying several mutations, including in the ETC cytochrome bo oxidase complex, e.g., *cyoB*. The findings further show that exposure to *cyoB* mutant *E. coli* leads to an iron deficiency and ATFS-1 mitochondrial stress sensor activation in the host intestinal cells, thus inducing neuronal activation. Induction of neuronal activity rescues iron deficiency and mitochondrial dysfunction in the intestine by activating ferritin expression.

We appreciate the presented findings showing a role of neuronal activation for resolving microbiota-induced iron deficiency in the *C. elegans* gut and realise that they will be of interest to the research field. At the same time, we noted that previous work has shown that neurons can regulate ferritin expression in the gut in response to oxygen levels. Furthermore, *cyoB* loss in *E. coli* has been shown to impair mitochondrial function in the *C. elegans* host via induction of bacterial ROS, thus leading to the host oxidative stress, and the potential impact of ROS generation has not been explored in your current study. Additionally, a recent paper from your team demonstrated the ability of *cyoB* mutant *E. coli* to activate host head neurons, albeit in order to induce an innate immune response. Finally, it is known that ATFS-1 senses mitochondrial stress and activates mitochondrial UPR, and that the latter can lead to non-cell autonomous effects on Wnt activation, such as demonstrated in your current study in the case of neuronal activation by intestinal ATFS-1. Taken together, I am afraid we concluded that the broader conceptual advance provided in your current manuscript is not sufficient to consider publication in The EMBO Journal.

Thank you for giving us the opportunity to consider this manuscript. I regret that I could not offer better news this time, and I sincerely hope for a rapid publication of this work at another venue.

Yours sincerely,

Ieva Gailite

** As a service to authors, EMBO Press provides authors with the possibility to transfer a manuscript that one journal cannot offer to publish to another EMBO publication or the open access journal Life Science Alliance launched in partnership between EMBO Press, Rockefeller University Press and Cold Spring Harbor Laboratory Press. The full manuscript and if applicable, reviewers' reports, are automatically sent to the receiving journal to allow for fast handling and a prompt decision on your manuscript. For more details of this service, and to transfer your manuscript please click on Link Not Available. **

Dear Ieva,

Thank you for your prompt response regarding the decision not to send our manuscript out for peer review. Based on your comments, I have realized that this decision may stem from an unclear presentation of the significance of our work in the cover letter. I am formally appealing this decision and would like to emphasize the major significance of our findings, which lie in the discovery that alterations in gut bacterial activity can change the levels of neurotransmitters in brain neurons. This is a highly important question that may shed light on how gut bacteria affect brain activity.

A point-by-point response to your comments:

1, We noted that previous work has shown that neurons can regulate ferritin expression in the gut in response to oxygen levels.

Response: We thank you for pointing this out. Indeed, neurons can regulate ferritin expression in the gut in response to oxygen levels. However, the major novelty of our work lies in the role of gut bacteria in triggering host brain neurotransmitter biosynthesis. It is not reasonable to expect every aspect of a study to be entirely novel, although the neurons involved in ferritin regulation in our study differ from those in previous work.

2, Furthermore, cyoB loss in E. coli has been shown to impair mitochondrial function in the C. elegans host via induction of bacterial ROS, thus leading to host oxidative stress. The potential impact of ROS generation has not been explored in your current study.

Response: We appreciate the editor's suggestion to investigate whether ROS plays a critical role in regulating bacteria-induced neuron activation. We have collected data showing that ROS generation does not play a significant role in bacteria-induced host neuron activation, as indicated by the lack of effect of antioxidant supplementation on bacteria-induced neuron activation. This data is not included in the current version of the manuscript but can be provided in a future version if required.

3, Additionally, a recent paper from your team demonstrated the ability of cyoB mutant E. coli to activate host head neurons, albeit to induce an innate immune response.

Response: Thank you for your attention to our recent publication, which reported the phenomenon that bacteria trigger the host innate immune response depending on the function of head neurons. However, I would like to highlight that the novelty of the manuscript submitted to EMBO J is not affected, since this paper focuses on the function and mechanism of gut bacteria in enhancing the biosynthesis of neurotransmitters in the brain, providing important mechanistic insights into the role of bacteria in affecting brain activity.

4, Finally, it is known that ATFS-1 senses mitochondrial stress and activates mitochondrial UPR, which can lead to non-cell autonomous effects on Wnt activation, as demonstrated in your current study in the case of neuronal activation by intestinal ATFS-1.

Response: Previous studies have found that neural activation of ATFS-1 can activate Wnt signaling, inducing mitochondrial UPR in the intestine. In our study, we found that intestinal ATFS-1 induces neuron activity by increasing neural Wnt (*cwn-2*) expression, which differs from the neuron-intestine communication reported in previous studies.

The profound impacts of our study are summarized as follows:

- A, We have uncovered that alterations in gut bacterial activity can enhance the biosynthesis of dopamine and serotonin in the head neurons of host *C. elegans*.
- B, We have revealed a surprising role of the brain in sensing peripheral iron levels.
- C, We have elucidated a previously unknown physiological role of head dopaminergic and serotonergic neurons in maintaining host iron homeostasis and mitochondrial function to cope with gut bacterial challenges.

Taken together, I kindly request that you reconsider the decision and send this exciting discovery out for peer review. A thorough evaluation by independent reviewers will provide a more comprehensive and objective assessment of the manuscript's quality and relevance. If you require any additional information or materials, please do not hesitate to contact me.

Sincerely,
Hongyun Tang

Dear Hongyun,

Thank you for submitting your manuscript for consideration by the EMBO Journal. We have now received comments from two reviewers, which are included below for your information. Since the third reviewer was not able to return their comments in a timely manner, I am taking the decision based on the input at hand.

As you will see, both reviewers are generally positive in their assessment and appreciate the contribution of the study to the research field. At the same time, they indicate a number of concerns that would be important to address in the revised study. From my side, I find these points generally reasonable. Therefore, I invite you to address these comments in a revised manuscript. I think that it would be useful to discuss the revision in more detail via email or phone/videoconferencing - please let me know which option you prefer.

We generally allow three months as standard revision time, which can be extended to six months in the case of major revisions. Should you foresee a problem in meeting this deadline, please let us know in advance to discuss an extension. As a matter of policy, competing manuscripts published during this period will not negatively impact on our assessment of the conceptual advance presented by your study. However, please contact me as soon as possible upon publication of any related work to discuss the appropriate course of action.

When preparing your letter of response to the referees' comments, please bear in mind that this will form part of the Review Process File and will therefore be available online to the community. For more details on our Transparent Editorial Process, please visit our website: <https://www.embopress.org/page/journal/14602075/authorguide#transparentprocess>. Please also see the attached instructions for further guidelines on preparation of the revised manuscript.

Please feel free to contact me if have any further questions regarding the revision. Thank you for the opportunity to consider your work for publication, and I look forward to discussing your revision with you.

With best wishes,

leva

leva Gailite, PhD
Senior Scientific Editor
The EMBO Journal
Meyerhofstrasse 1
D-69117 Heidelberg
Tel: +4962218891309
i.gailite@embojournal.org

- a point-by-point response to the referees' comments, with a detailed description of the changes made (as a word file).
- a word file of the manuscript text.
- individual production quality figure files (one file per figure)
- a complete author checklist, which you can download from our author guidelines

(<https://www.embopress.org/page/journal/14602075/authorguide>).

- Expanded View files (replacing Supplementary Information)

- a Reagents and Tools Table as part of the Methods section, which can be downloaded from our author guidelines

(<https://www.embopress.org/page/journal/14602075/authorguide#structuredmethods>)

We realize that it is difficult to revise to a specific deadline. In the interest of protecting the conceptual advance provided by the work, we recommend a revision within 3 months (21st May 2025). Please discuss the revision progress ahead of this time with the editor if you require more time to complete the revisions.

Referee #1:

Li, Tang and colleagues report a role for aminergic neurons in sensing and responding to altered iron metabolism in *C. elegans*. The central hypothesis - iron deficiency is sensed by the gut and relayed to neurons that feed back to the gut - is interesting and supported by multiple lines of experimentation. The study, however, does have several gaps and some points require clarification or technical controls. I list these below. Overall, the study is clearly presented and, in my opinion, constitutes an advance in our understanding of how the nervous system regulates metabolism and whole-organism physiology.

1. It is not clear whether changes in reporter gene expression are caused by changes in transcription, translation, or both. The authors should clarify this - their model predicts that iron availability regulates transcription of neuronal genes.
2. The authors do not report the results of their screen of *E. coli* mutants and only identify the subset that function together in bacterial respiration. The other genes identified by their screen should be listed in a table. These data provide important context for the reader.
3. The authors must show source data for Fig. 1F along with appropriate controls indicating that the chemical species they identify as serotonin and dopamine have been correctly identified.
4. The authors should demonstrate that the increased calcein AM signal is not a consequence of altered permeability of animals to the dye. There exist many AM-modified dyes, and the authors should demonstrate that exposure to mutant *E. coli* does not alter the signal of a dye that is not sensitive to free iron.
5. The effect of *bas-1* mutation on free iron levels shown in Fig. 3A seems much more dramatic than the quantification shown in Fig. 3B. The authors should confirm that the quantification is correct and that the representative micrographs are indeed representative.
6. Figure EV3 shows that loss of a vesicular monoamine transporter also affects free iron availability. The authors should compare the effect size of *bas-1* mutation to that of *cat-1* mutation. Their model predicts that these two mutations should have similar effects on free iron.
7. The authors' model further predicts that *bas-1*; *cat-1* double mutants will have defects similar to that of the single mutants. By contrast, *cat-2*; *tph-1* double mutants should have a stronger defect than either single mutant.
8. The authors do not capitalize on all the genetic tools available to investigate biogenic amine signaling in *C. elegans*. Mutation of *tdc-1* eliminates tyramine and octopamine, and *cat-4* mutation causes defects similar to those caused by *bas-1* mutation. The authors should consider testing these mutants for defects in regulation of free iron.
9. The effect of *bas-1* mutation on oxygen consumption should be validated with appropriate controls. The authors could, for example, test whether the effect is reduced by exogenous neurotransmitters. Alternately, the authors could use rescuing transgenes to show that the effect on oxygen consumption is indeed caused by the *bas-1* mutation and not by a background mutation.
10. The table shown in Figure EV6A lacks data and is impossible to interpret. The authors should show the data that lead to the conclusions shown in this table.
11. The authors should consider determining the timecourse of BAS-1 induction by reduced free iron. Can this effect be seen within a few hours?
12. Tools exist to manipulate neural activity in a targeted manner. The authors should consider testing whether silencing electrical activity in aminergic neurons reproduces the effect seen by mutating amine synthesis pathways. At least the authors should discuss what role, if any, neural activity might have in this phenomenon.

13. The authors should discuss more concretely how altered iron metabolism might affect behavior. Aminergic neurons in *C. elegans* have well defined roles in controlling behavior, including foraging behavior. The discussion of how this gut-brain axis might impact behavior is limited to speculation about how a vertebrate counterpart might function. There are obvious implications for *C. elegans* behavior that should be explored.

Minor comment

1. Micrographs showing green fluorescence are labeled 'FITC' throughout the figures. This is confusing as FITC is never the dye used for fluorescence microscopy.

Referee #2:

Gut microbes play an important role in health and disease, particularly in regulating brain functions. In this manuscript, using *C. elegans* as a model, the authors highlight an intriguing feedback model - 'gut-brain-gut axis' - to demonstrate neuronal regulation of iron homeostatic imbalance caused by gut microbial change. Mechanistically, they discovered most of the key events/factors and straightened out the main pathways. While most published work touches on gut-to-brain or brain-to-gut signaling, this paper investigated gut-to-brain signaling, then back to the gut. This is an interesting conceptual point distinguishing this work from those in literature.

Overall, this is a nice piece of work. The experimental data mostly supports the conclusions. I am thus happy to support its publication in EMBOJ. Nevertheless, several issues need to be addressed before it is considered for publication.

1, Are there any *E. coli* mutants that lead to a decrease in BAS-1 expression? Or is there any particular reason for the authors to select the mutants that increase BAS-1 expression?

2, How does Δ cyoB *E. coli* cause a reduction of labile iron in intestine cells of *C. elegans*? Does Δ cyoB *E. coli* intake and store more iron than BW25113?

3, Does supplementation of ferrous iron work the same as ferric ion? Is dietary *E. coli* required for worms to absorb and/or store ferrous/ferric iron?

4, Any NLPs or other peptides released from intestinal cells function as cell-nonautonomous signaling factors?

5, There are two ferritin proteins in *C. elegans*, FTN-1 and FTN-2. Do Δ cyoB *E. coli* and head dopaminergic/serotonergic neurons change the protein level of FTN-1 or FTN-2?

Referee #1:

Li, Tang and colleagues report a role for aminergic neurons in sensing and responding to altered iron metabolism in *C. elegans*. The central hypothesis - iron deficiency is sensed by the gut and relayed to neurons that feed back to the gut - is interesting and supported by multiple lines of experimentation. The study, however, does have several gaps and some points require clarification or technical controls. I list these below. Overall, the study is clearly presented and, in my opinion, constitutes an advance in our understanding of how the nervous system regulates metabolism and whole-organism physiology.

Response: We sincerely appreciate the reviewer's thoughtful evaluation of our manuscript and their positive assessment of our work. We have re-examined the entire paper to improve clarity and have incorporated additional data to strengthen the study in response to the reviewer's constructive feedback.

1. It is not clear whether changes in reporter gene expression are caused by changes in transcription, translation, or both. The authors should clarify this - their model predicts that iron availability regulates transcription of neuronal genes.

Response: We appreciate the reviewer's query regarding the potential contribution of transcription to the induced *Is[bas-1p::BAS-1::GFP]* reporter signal. We conducted qPCR analyses, which confirmed that the mRNA level of *bas-1* increased after $\Delta cyoB$ *E. coli* treatment compared to wild-type BW25113 *E. coli* treatment. This finding supports that the increased expression of the *Is[bas-1p::BAS-1::GFP]* reporter signal is at least partially attributed to a transcriptional increase. We have incorporated these new data into the revised manuscript (lines 167-169, Fig. EV1E). We are grateful for the reviewer's suggestions, which have significantly strengthened our conclusions.

2. The authors do not report the results of their screen of *E. coli* mutants and only identify the subset that function together in bacterial respiration. The other genes identified by their screen should be listed in a table. These data provide important context for the reader.

Response: We thank the reviewer for this valuable suggestion. In the revised manuscript, we have included Table EV1, which catalogs all 29 *E. coli* mutants identified in our genome-wide screen (3,985 mutants tested) that significantly increased neuronal *Is[bas-1p::BAS-1::GFP]* expression. The table provides:

- Gene names
- Locus
- % of increase in the *Is[bas-1p::BAS-1::GFP]* transgene expression
- Statistical significance (p-values)
- Sample size (n) for each mutant
- Gene description

We appreciate the reviewer's input, which strengthens the manuscript's utility for future studies of gut-brain interactions.

3. The authors must show source data for Fig. 1F along with appropriate controls

indicating that the chemical species they identify as serotonin and dopamine have been correctly identified.

Response: We thank the reviewer for this constructive comment and apologize for this omission. In the revised manuscript, we have included the raw mass spectrometry data for serotonin and dopamine quantification (Fig. EV1F-K), using Sigma-Aldrich standards (purity $\geq 98\%$) for validation. Compounds were confirmed via:

- Retention time alignment (± 0.1 min)
- Precursor/product ion matches (serotonin: m/z 177.1 \rightarrow 159.9; dopamine: m/z 154.1 \rightarrow 90.9)
- Consistent elution profiles with standards

These criteria validate our identification of serotonin and dopamine in *C. elegans* samples. We appreciate the reviewer's input, which improves our manuscript.

4. The authors should demonstrate that the increased calcein AM signal is not a consequence of altered permeability of animals to the dye. There exist many AM-modified dyes, and the authors should demonstrate that exposure to mutant *E. coli* does not alter the signal of a dye that is not sensitive to free iron.

Response: We thank the reviewer's constructive suggestions. To confirm that elevated Calcein-AM signals reflect a decrease in labile iron rather than nonspecific cell permeability changes, we followed the reviewer's advice and conducted parallel experiments using Fluo-4 AM, an AM ester dye with identical cellular uptake and hydrolysis pathways to Calcein-AM but sensitive to Ca^{2+} (not iron). While the *cyoB* mutant *E. coli* robustly increased Calcein-AM fluorescence in the worms (Fig. 2A), Fluo-4 AM signals remained unchanged compared to wild-type BW25113 controls (Fig. EV2A). This validation confirms that Calcein-AM specifically reports iron status, not permeability artifacts in our experiments. We have incorporated these results into the revised manuscript (lines 200-204) and again thank the reviewer for strengthening our work.

5. The effect of *bas-1* mutation on free iron levels shown in Fig. 3A seems much more dramatic than the quantification shown in Fig. 3B. The authors should confirm that the quantification is correct and that the representative micrographs are indeed representative.

Response: We thank the reviewer for the careful evaluation of our data. When preparing our original paper, we shared the same impression as the reviewer and were therefore very careful in our quantification to ensure precision. To ensure objective quantification of Calcein fluorescence, we applied uniform ImageJ thresholds across all groups, measuring mean intensity (not total fluorescence) to avoid bias from variable signal area. One possible explanation for the difference in perception between visual inspection and ImageJ quantification is that our software quantification measures mean fluorescence intensity, while visual perception may be influenced by both intensity and area (total fluorescence), as well as other factors.

We have double-checked all the raw data, confirming that the quantification is correct. Here, we also provide all the source images of Calcein-AM staining of N2

worms treated with *cyoB* mutant bacteria (Figure 1 in the response letter) and those of *bas-1(lf)* worms treated with *cyoB* mutant bacteria (Figure 2 in the response letter) to indicate that the micrographs in Fig. 3A are indeed representative. We appreciate the reviewer's diligence.

Figure for reviewers removed.

Figure for reviewers removed.

6. Figure EV3 shows that loss of a vesicular monoamine transporter also affects free iron availability. The authors should compare the effect size of *bas-1* mutation to that of *cat-1* mutation. Their model predicts that these two mutations should have similar effects on free iron.

Response: We thank the reviewer for the insightful comments. Indeed, our data reveal that *bas-1(lf)* mutants, which are defective in serotonin and dopamine synthesis, exhibit more severe iron depletion than *cat-1(lf)* mutants, which are impaired in monoamine vesicular transport, under $\Delta cyoB$ *E.coli* challenge (Fig. EV3D). This divergence likely arises because CAT-1 transports various monoamines (Duerr, Frisby et al., 1999), including octopamine and tyramine, which are synthesized independently of BAS-1. Consistent with this, with the reviewer's suggestion, we found that *tdc-1* mutation, which eliminates octopamine and tyramine (Alkema, Hunter-Ensor et al., 2005), rescues $\Delta cyoB$ -induced iron loss (Appendix Fig. S2), opposing the iron-depleting effects of losing dopamine and serotonin. This antagonism explains why *cat-1(lf)*, which disrupts both iron-promoting (dopamine and serotonin) and iron-limiting (octopamine and tyramine) signals, has a milder phenotype than *bas-1(lf)*.

Furthermore, *tph-1(lf)* (serotonin-deficient) or *cat-2(lf)* (dopamine-deficient) single mutants show less iron depletion than *bas-1(lf)* (Fig. EV3). However, our newly added data demonstrate that the *tph-1(lf); cat-2(lf)* double mutant exhibits a

560% increase in the Calcein signal compared to the wildtype worm control (Fig. EV3A). Similarly, the Calcein signal is enhanced by 550% in the *bas-1(lf)* mutant compared to the wildtype worm control (Fig. EV3D). This further confirms that serotonin and dopamine act together to counteract bacterial iron deficiency.

Therefore, our original and these newly added data collectively observe a dual regulatory axis: serotonin and dopamine enhance iron availability, while octopamine and tyramine may suppress it. We again thank the reviewer for their input, which has strengthened our conclusions.

7. The authors' model further predicts that *bas-1*; *cat-1* double mutants will have defects similar to that of the single mutants. By contrast, *cat-2*; *tph-1* double mutants should have a stronger defect than either single mutant.

Response: We thank the reviewer for this excellent point regarding the genetic interactions between these mutations. Indeed, following the reviewer's suggestion, we observed that the *cat-2(lf);tph-1(lf)* double mutant displays a more severe iron reduction than either single mutant alone (Fig. EV3A in the new manuscript) and *bas-1(lf); cat-1(lf)* double mutants exhibit defects similar to those of the *bas-1(lf)* single mutant (Fig. EV3D in the new manuscript). As mentioned above, *cat-1* not only affects the release of dopamine and serotonin but also influences the release of other monoamine transmitters. Therefore, the genetic interaction between *bas-1(lf)* and *cat-1(lf)* could be more complex due to their shared and divergent roles in neuronal regulation.

Notably, following the reviewer's suggestion, we collect new data indicating that the *tph-1(lf); cat-2(lf)* double mutant exhibits a 560% increase in the Calcein signal compared to the wildtype worm control (Fig. EV3A). Similarly, the Calcein signal is enhanced by 550% in the *bas-1(lf)* mutant compared to the wildtype worm control (Fig. EV3D). These findings, combined with the observation that the *cat-2(lf);tph-1(lf)* double mutant displays a more severe iron deficiency than either single mutant alone (Fig. EV3A in the new manuscript), further supports our original conclusion that dopamine and serotonin signalings act together to promote iron availability. All these new data have been included in the revised manuscript and we appreciate the reviewer's suggestions, which have strengthened our conclusion.

8. The authors do not capitalize on all the genetic tools available to investigate biogenic amine signaling in *C. elegans*. Mutation of *tdc-1* eliminates tyramine and octopamine, and *cat-4* mutation causes defects similar to those caused by *bas-1* mutation. The authors should consider testing these mutants for defects in regulation of free iron.

Response: We thank the reviewer for highlighting these valuable genetic tools to further probe biogenic amine signaling in iron regulation. Following the reviewer's suggestion, we have conducted new experiments with both *cat-4(lf)* and *tdc-1(lf)* mutants. Consistent with the reviewer's prediction and our conclusion, the *cat-4(e1141)* mutant exhibits similar levels of iron availability reduction as observed in

the *bas-1(lf)* mutation in the presence of *cyoB* mutant *E. coli* (Fig. EV3E in the new manuscript). Interestingly, the *cat-4(e1141)* mutant shows a reduction in iron availability even in the presence of wild-type bacteria (Fig. EV3E in the new manuscript), which may be due to CAT-4's involvement in additional processes (Loer, Calvo et al., 2015), suggesting an intriguing direction for future exploration. These data support a critical role of *cat-4* in maintaining iron homeostasis.

Following the reviewer's suggestions, we also performed new experiments with the *tdc-1(n3420)* mutation as mentioned above and found that eliminating tyramine and octopamine by *tdc-1* mutation can alleviate labile iron reduction caused by *cyoB* mutant bacteria. These results reveal a previously-unknown role of tyramine and octopamine in negatively regulating free iron levels, which represents an important direction for future work. Based on these observations, a new discussion has been included in the new manuscript (lines 681-688) as follows: "In addition to insufficiency, excessive body iron also causes severe pathological consequences (Wang & Babitt, 2019), suggesting the existence of neurons for negative regulation of iron levels. Indeed, we found that neurons may use tyramine- and octopamine-mediated signaling to negatively regulate iron levels (Appendix Fig. S2), as eliminating tyramine and octopamine through *tdc-1(lf)* mutation alleviates the labile iron reduction caused by *cyoB* mutant bacteria. The presence of both positive and negative neuronal regulations of iron levels aligns with the evolutionary need for precise iron control." We appreciate the reviewer's suggestions, which have strengthened our conclusion.

9. The effect of *bas-1* mutation on oxygen consumption should be validated with appropriate controls. The authors could, for example, test whether the effect is reduced by exogenous neurotransmitters. Alternately, the authors could use rescuing transgenes to show that the effect on oxygen consumption is indeed caused by the *bas-1* mutation and not by a background mutation.

Response: We appreciate the reviewer's suggestion. Following this advice, we conducted new experiments with exogenous serotonin supplementation and found that indeed, this treatment significantly reversed the reduced OCR phenotype in the *bas-1(lf)* mutant in the presence of *cyoB* mutant bacteria (Figure EV4C in the new manuscript). These findings support the conclusion that the decreased oxygen consumption in the *bas-1* mutant is due to the *bas-1* mutation rather than a background mutation. Additionally, exogenous serotonin supplementation increased OCR even in wild-type worms treated with *cyoB* mutant bacteria (Figure EV4C in the new manuscript), further substantiating the role of serotonin in promoting iron availability, as reported in our original manuscript (Figure 3H). We are grateful for the reviewer's suggestions, which have strengthened our conclusions.

10. The table shown in Figure EV6A lacks data and is impossible to interpret. The authors should show the data that lead to the conclusions shown in this table.

Response: We sincerely thank the reviewer for this constructive comment. We have now replaced the original Figure EV6A with six bar graphs that more clearly illustrate

the statistical data (Fig. EV6 A-F in the revised manuscript). We appreciate the reviewer's suggestion for enhancing the clarity of our data presentation.

11. The authors should consider determining the timecourse of BAS-1 induction by reduced free iron. Can this effect be seen within a few hours?

Response: We appreciate the reviewer's insightful suggestion. In accordance with the reviewer's recommendation, we conducted time-course experiments to assay BAS-1::GFP expression in L4-stage worms treated with either BW25113 (control) or $\Delta cyoB$ *E. coli* at 3, 6, 12, and 24 hours post-exposure. The results revealed that BAS-1 upregulation was detectable only after 24 hours of exposure to $\Delta cyoB$ *E. coli*, with no significant changes at earlier time points (3–12 hours) (Figure 3 in the response letter). This slow response suggests that bacterial modulation of host neuronal gene expression is a chronic process rather than an acute response. We thank the reviewer for the suggestion, which has provided new insights into our understanding of the bacteria-neuron interaction process.

Figure 3: quantitative analyses of BAS-1::GFP expression in L4 *C. elegans* treated with BW25113 or *cyoB* mutant bacteria were conducted at 3, 6, 12, and 24 hours. For each group, $n > 30$. Statistical significance was determined by unpaired t-test, with ns indicating $p > 0.05$ and * indicating $p < 0.05$.

12. Tools exist to manipulate neural activity in a targeted manner. The authors should consider testing whether silencing electrical activity in aminergic neurons reproduces the effect seen by mutating amine synthesis pathways. At least the authors should discuss what role, if any, neural activity might have in this phenomenon.

Response: We thank the reviewer for this insightful suggestion. We tested whether enhancing neuronal electrical excitability could promote labile iron levels by analyzing *goa-1(sa734)* mutants, which lack the $G\alpha_o$ subunit necessary for hyperpolarizing neurons, including serotonergic neurons, thereby reducing their electrical excitability (Ravi, Zhao et al., 2021). Since the loss of $G\alpha_o$ potentiates neuronal activity and serotonin release, this mutation should mimic the effect of exogenous serotonin supplementation on promoting labile iron. Indeed, *goa-1(sa734)* suppressed $\Delta cyoB$ -induced labile iron decrease (Appendix Fig. S4). These results

suggest that enhanced serotonergic neuron electrical activity counteracts bacteria-induced iron decrease, maybe via increasing neurotransmitter release.

In the revised manuscript (lines 688-696), we now discuss the implications of electrical activity in iron regulation: “Moreover, our findings not only establish BAS-1-dependent neurotransmitter synthesis as an important mechanism for iron regulation, but also imply the involvement of neuronal electrical activity in modulating peripheral iron metabolism. For example, the *goa-1(sa734)* mutation, which lacks the $G\alpha_o$ subunit and potentiates serotonergic neuronal electrical excitability and serotonin release (Ravi et al., 2021), reverses the *cyoB*-induced iron decrease (Appendix Fig. S4), as that observed with exogenous serotonin supplementation. This implies that neuronal electrical activity may modulate iron homeostasis by governing neurotransmitter secretion, adding another layer of neuronal regulation of iron metabolism.”

We thank the reviewer for these suggestions, which have enhanced our manuscript and prompted new research directions.

13. The authors should discuss more concretely how altered iron metabolism might affect behavior. Aminergic neurons in *C. elegans* have well defined roles in controlling behavior, including foraging behavior. The discussion of how this gut-brain axis might impact behavior is limited to speculation about how a vertebrate counterpart might function. There are obvious implications for *C. elegans* behavior that should be explored.

Response: We thank the reviewer for the comments regarding our discussion of the link between iron metabolism and behavior. In response, we have conducted new experiments. Our data reveal that wild-type *C. elegans* avoid $\Delta cyoB$ bacteria (iron-deficient) in favor of BW25113 wildtype bacteria (Appendix Fig. S5). This avoidance is reversed by iron supplementation, confirming that iron depletion, rather than other $\Delta cyoB$ effects, drives foraging preferences. Consistent with this, *bas-1(lf)* mutants (exacerbated iron deficiency; Fig. 3A) exhibit a stronger aversion to $\Delta cyoB$ bacteria than wild-type worms (Appendix Fig. S5), which is also reversed by iron repletion (Appendix Fig. S5). These findings support that iron deficiency in the animals motivates their avoidance of iron-poor food sources.

In the revised manuscript (lines 728-737), we have followed the reviewer's suggestion to improve the discussion with our new data: " Moreover, our findings may stimulate new studies to investigate how brain defects could cause iron imbalance as well as whether iron insufficiency alters brain activity in other animals. For example, an obvious symptom of iron deficiency anemia in human is the mysterious pica, which is an unusual eating habit or craving for nonnutritive substances (Coltman, 1969), suggesting that iron insufficiency alters brain activity to modify behavior. In support of this idea, we found that iron deficiency drives worms to seek food with higher iron content (Appendix Fig. S5), underscoring the critical role of dopaminergic and serotonergic neurons in linking iron metabolism with foraging behaviors (D'Agostino, Lyons et al., 2018, Hills, Brockie et al., 2004). "

We again thank the reviewer for their input, which has enhanced our manuscript

and highlighted a key future direction.

Minor comment

1. Micrographs showing green fluorescence are labeled 'FITC' throughout the figures. This is confusing as FITC is never the dye used for fluorescence microscopy.

Response: We sincerely appreciate the reviewer's careful review of our data. In the revised manuscript, we have replaced "FITC" with appropriate labeling, including Calcein-AM, *hsp-6p::GFP*, *ftn-1p::GFP*, and BAS-1::GFP, in all relevant figures to accurately reflect the specific dyes or reporter genes used in our experiments. We are grateful for the reviewer's suggestion, which has significantly enhanced the precision and clarity of our manuscript.

Referee #2:

Gut microbes play an important role in health and disease, particularly in regulating brain functions. In this manuscript, using *C. elegans* as a model, the authors highlight an intriguing feedback model - 'gut-brain-gut axis' - to demonstrate neuronal regulation of iron homeostatic imbalance caused by gut microbial change.

Mechanistically, they discovered most of the key events/factors and straightened out the main pathways. While most published work touches on gut-to-brain or brain-to-gut signaling, this paper investigated gut-to-brain signaling, then back to the gut. This is an interesting conceptual point distinguishing this work from those in literature.

Overall, this is a nice piece of work. The experimental data mostly supports the conclusions. I am thus happy to support its publication in EMBOJ. Nevertheless, several issues need to be addressed before it is considered for publication.

Response: We sincerely thank the reviewer for their positive evaluation of our work, particularly the novelty of our discovery of the "gut-brain-gut axis" in regulating iron homeostasis, and for their insightful suggestions to further enhance the manuscript. We have incorporated new data and enhanced descriptions as suggested by the reviewer to improve our study.

1, Are there any *E. coli* mutants that lead to a decrease in BAS-1 expression? Or is there any particular reason for the authors to select the mutants that increase BAS-1 expression?

Response: We appreciate the reviewer's thoughtful question. In this study, we focused on *E. coli* mutants that enhance BAS-1 expression because dopamine and serotonin, synthesized via BAS-1, are crucial neurotransmitters whose deficiency is associated with various neurological disorders. Identifying bacterial strains that upregulate BAS-1, and thus neurotransmitter production, could inform strategies for intervening in conditions linked to dopamine and serotonin deficiency. In response to the reviewer's comments, we have included this point in the Discussion section of the revised manuscript (lines 638-641).

During our screen of 3,985 *E. coli* mutants, we indeed observed some mutants that triggered a decrease in BAS-1::GFP expression in the host. However, as our primary objective was to identify *E. coli* mutants that could increase BAS-1::GFP expression in the worm, these mutants were not pursued further with additional biological replicates. We agree with the reviewer that investigating microbes capable of suppressing neurotransmitter production would be valuable for understanding neurological dysfunction, representing an intriguing direction for future research. We thank the reviewer for these comments, which have improved our manuscript.

2, How does Δ cyoB *E. coli* cause a reduction of labile iron in intestine cells of *C. elegans*? Does Δ cyoB *E. coli* intake and store more iron than BW25113?

Response: We sincerely appreciate this insightful question regarding the mechanism by which Δ cyoB *E. coli* causes a reduction in labile iron in intestinal cells of *C. elegans*. Following the reviewer's suggestion, we tested the total iron levels in bacteria using Inductively Coupled Plasma-Mass Spectrometry (ICP-MS) (Kaur,

Yantiri et al., 2003). The *fur* mutant bacteria, which block iron transport, exhibited decreased iron levels, validating the effectiveness of our assay. In contrast, the *cyoB* mutant bacteria did not show significant changes in total iron levels compared to the wild-type BW25113 bacteria (Fig. 4 in this response letter). This data indicates that the host iron deficiency observed with $\Delta cyoB$ *E. coli* is not due to increased iron intake and storage by these bacteria.

This intriguing observation suggests that $\Delta cyoB$ may induce host iron deficiency through other unknown mechanisms, which represents an important future research direction. We thank the reviewer for their constructive suggestion, which has inspired new research directions for future study.

Figure for reviewers removed.

3, Does supplementation of ferrous iron work the same as ferric iron? Is dietary *E. coli* required for worms to absorb and/or store ferrous/ferric iron?

Response: We greatly appreciate the reviewer's insightful questions. Previous studies have demonstrated that dietary Fe^{3+} is reduced to Fe^{2+} at the intestinal surface by ferric reductases before cellular uptake (Gunshin, Mackenzie et al., 1997). Therefore, supplementation with ferrous iron should have the same effect as ferric iron. To address the reviewer's question, we performed supplementation with 4 mM ferrous sulfate ($FeSO_4$) in the medium. Indeed, similar to the effect of ferric iron supplementation, our results show that ferrous (Fe^{2+}) iron supplementation also suppressed intestinal iron decrease as well as neuronal BAS-1 activation induced by *cyoB* mutant bacteria. These new data are included in the revised manuscript (Appendix Fig. S1).

Notably, worms can be grown in synthetic medium without bacteria (Samuel, Sinclair et al., 2014), suggesting that dietary *E. coli* may not be essential for worms to absorb and/or store ferrous or ferric iron. Combining our results, which show no

significant decrease in total iron content in the *cyoB* mutant compared to the BW wild-type *E. coli* (Figure 4 in the response letter), it is unlikely that the *cyoB* mutant *E. coli* affects the iron metabolism of the worm by merely acting as a source of dietary iron. We appreciate the reviewer's constructive comments, which have significantly improved our manuscript.

4, Any NLPs or other peptides released from intestinal cells function as cell-nonautonomous signaling factors?

Response: We appreciate the reviewer's insightful question regarding potential neuropeptide signals mediating gut-to-brain communication. To test the possibility of neuropeptide-mediated gut-brain signaling, we conducted a systematic investigation combining transcriptomic screening and functional validation. Specifically, RNA-seq analysis of *C. elegans* treated with four different conditions—BW25113, *cyoB* mutant bacteria, BW25113 supplemented with ferric iron, and *cyoB* mutant bacteria supplemented with ferric iron—identified 14 iron-responsive neuropeptide candidates (NLPs, FLPs, INS-family) upregulated by at least 2-fold in the *cyoB* mutant bacteria treatment group and reversed by iron supplementation. Intestine-specific RNAi knockdown of these peptides, however, did not affect *cyoB*-induced neuronal BAS-1::GFP activation (Appendix Fig. S3), suggesting that none of these single peptides is essential for gut-to-brain cell-nonautonomous signaling.

These results imply redundancy among neuropeptide families or alternative mechanisms mediating gut-brain communication, highlighting the complexity of this interaction and the need for further exploration in future studies. In response to the reviewer's comments, we have included this point in the Discussion of the revised manuscript (lines 674-680). We thank the reviewer for their constructive comments, which have improved our manuscript and deepened our understanding of gut-brain communication.

5, There are two ferritin proteins in *C. elegans*, FTN-1 and FTN-2. Do $\Delta cyoB$ *E. coli* and head dopaminergic/serotonergic neurons change the protein level of FTN-1 or FTN-2?

Response: We thank the reviewer for this constructive question regarding the levels of the two ferritin proteins, FTN-1 and FTN-2, in *C. elegans*. To address the reviewer's query, we employed LC-MS/MS to quantify the protein levels of FTN-1 and FTN-2, as previously reported (Hughes, Moggridge et al., 2019). Our quantitative LC-MS/MS analysis revealed a significant increase in FTN-1 protein levels in worms treated with $\Delta cyoB$ *E. coli* compared to those treated with BW25113 controls (Figure EV7D in the revised manuscript), aligning with our observation of increased *ftn-1* expression in worms treated with $\Delta cyoB$ *E. coli* in our original manuscript. In contrast, no significant change in FTN-2 protein levels was detected in worms treated with $\Delta cyoB$ *E. coli* compared to those treated with BW25113 wildtype *E. coli* (Figure EV7E in the revised manuscript), suggesting that *ftn-2* may not be involved in the neuronal regulation of iron levels in the context of $\Delta cyoB$ *E. coli* treatment.

We have incorporated these new data on FTN-1 and FTN-2 protein levels into

the revised manuscript (Fig. EV7D and EV7E), which further support the role of FTN-1 as a critical regulator in the neuronal modulation of iron availability in worms treated with *ΔcyoB E. coli*. We are grateful to the reviewer for their valuable feedback, which has contributed to the enhancement of our research.

References:

- Alkema MJ, Hunter-Ensor M, Ringstad N, Horvitz HR (2005) Tyramine Functions independently of octopamine in the *Caenorhabditis elegans* nervous system. *Neuron* 46: 247-60
- Coltman CA, Jr. (1969) Pagophagia and iron lack. *JAMA* 207: 513-6
- D'Agostino G, Lyons D, Cristiano C, Lettieri M, Olarte-Sanchez C, Burke LK, Greenwald-Yarnell M, Cansell C, Doslikova B, Georgescu T, Martinez de Morentin PB, Myers MG, Jr., Rochford JJ, Heisler LK (2018) Nucleus of the Solitary Tract Serotonin 5-HT_{2C} Receptors Modulate Food Intake. *Cell Metab* 28: 619-630 e5
- Duerr JS, Frisby DL, Gaskin J, Duke A, Asermely K, Huddleston D, Eiden LE, Rand JB (1999) The *cat-1* gene of *Caenorhabditis elegans* encodes a vesicular monoamine transporter required for specific monoamine-dependent behaviors. *J Neurosci* 19: 72-84
- Gunshin H, Mackenzie B, Berger UV, Gunshin Y, Romero MF, Boron WF, Nussberger S, Gollan JL, Hediger MA (1997) Cloning and characterization of a mammalian proton-coupled metal-ion transporter. *Nature* 388: 482-8
- Hills T, Brockie PJ, Maricq AV (2004) Dopamine and glutamate control area-restricted search behavior in *Caenorhabditis elegans*. *J Neurosci* 24: 1217-25
- Hughes CS, Moggridge S, Muller T, Sorensen PH, Morin GB, Krijgsveld J (2019) Single-pot, solid-phase-enhanced sample preparation for proteomics experiments. *Nat Protoc* 14: 68-85
- Kaur D, Yantiri F, Rajagopalan S, Kumar J, Mo JQ, Boonplueang R, Viswanath V, Jacobs R, Yang L, Beal MF, DiMonte D, Volitaskis I, Ellerby L, Cherny RA, Bush AI, Andersen JK (2003) Genetic or pharmacological iron chelation prevents MPTP-induced neurotoxicity in vivo: a novel therapy for Parkinson's disease. *Neuron* 37: 899-909
- Loer CM, Calvo AC, Watschinger K, Werner-Felmayer G, O'Rourke D, Stroud D, Tong A, Gotenstein JR, Chisholm AD, Hodgkin J, Werner ER, Martinez A (2015) Cuticle integrity and biogenic amine synthesis in *Caenorhabditis elegans* require the cofactor tetrahydrobiopterin (BH₄). *Genetics* 200: 237-53
- Ravi B, Zhao J, Chaudhry SI, Signorelli R, Bartole M, Kopchok RJ, Guijarro C, Kaplan JM, Kang L, Collins KM (2021) Presynaptic Galphao (GOA-1) signals to depress command neuron excitability and allow stretch-dependent modulation of egg laying in *Caenorhabditis elegans*. *Genetics* 218
- Samuel TK, Sinclair JW, Pinter KL, Hamza I (2014) Culturing *Caenorhabditis elegans* in axenic liquid media and creation of transgenic worms by microparticle bombardment. *J Vis Exp*: e51796

Wang CY, Babitt JL (2019) Liver iron sensing and body iron homeostasis. *Blood* 133: 18-29

Dear Hongyun,

Thank you for submitting a revised version of your manuscript. It has now been seen by two of the original reviewers, and I have copied their comments below. As you can see, they are generally satisfied with the revisions and now recommend acceptance of the manuscript after incorporation of textual revisions as requested by reviewer #1.

Additionally, there are a few formatting aspects that need to be addressed before I can extend official acceptance of the manuscript:

1. Please check that the funding information is correct and identical both in the manuscript and our online system. Currently, Westlake Education Foundation of Westlake University and the Zhejiang Provincial Key Laboratory Construction Project are missing in our online system.
2. CRediT has replaced the traditional author contributions section because it offers a systematic, machine-readable author contributions format that allows for more effective research assessment. Please remove the Authors Contributions from the manuscript and use the free text boxes beneath each contributing author's name in our online submission system to add specific details on the author's contribution. More information is available in our guide to authors.
3. Please rename "Declaration of interests" section into "Disclosure and competing interests statement" (further info: <https://www.embopress.org/page/journal/14602075/authorguide#conflictsofinterest>)
4. Please update references according to The EMBO Journal style - where there are more than 10 authors on a paper, the first 10 should be listed, followed by 'et al.' Please see further information here: <https://www.embopress.org/page/journal/14602075/authorguide#referencesformat>
5. When submitting your revised manuscript, please do not include the Reagents and Tools Table in the Methods section of the manuscript but upload it as a separate file choosing the file type "Reagent Table".
6. We require a "Data Availability" section at the end of " Methods". As far as I can see, no data deposition in external databases is needed for this paper. If I am correct, then please state in this section: "This study includes no data deposited in external repositories". Further information can be found at <https://www.embopress.org/page/journal/14602075/authorguide#dataavailability>
7. In the Appendix, please add page numbers to the table of contents.
8. In our standard image integrity check, we noticed reuse of figure panels between figures 2A and EV3B. If this intentional, please clearly indicate this in figure legends.
9. Please upload the updated source data files in the system and add an explanation on how image data were analysed in the Methods section, including the information on the data normalisation procedure. It might be useful to also add a summary of the applied signal normalisation approach to the appropriate source data files/folders.
10. Our data editors have flagged the following issues in figure legends that need correcting:
 - Please provide the exact p values in the legends of figures 1C, E, F, G; 2B, D, F; 3B, D, F, H; 4A, C, E, G; 5C, E, G, I; 6A-F; 7A, D, E, F; EV1 E, EV2 B, EV3 A, C, D, E; EV4 C, D; EV5 B, EV6 A-G; EV7 A, B, C, D, G, H; S1 A, B; S2, S3, S4, S5.
 - Please define the error bars in the legend of figure S3.
11. Papers published in The EMBO Journal are accompanied online by a 'Synopsis' to enhance discoverability of the manuscript. It consists of A) a short (1-2 sentences) summary of the findings and their significance, B) 3-4 bullet points highlighting key results and C) a synopsis image that is 550x300-600 pixels large (width x height, jpeg or png format). You can either show a model or key data in the synopsis image. Please note that the image size is rather small and that text needs to be readable at the final size.

With best wishes,

Ieva

We realize that it is difficult to revise to a specific deadline. In the interest of protecting the conceptual advance provided by the work, we recommend a revision within 3 months (24th Dec 2025). Please discuss the revision progress ahead of this time with the editor if you require more time to complete the revisions.

Referee #1:

The revised manuscript of Li, Tong and colleagues includes substantial amounts of new data that address comments raised during review of the initial submission. Important controls and cross-checks of key conclusions are included in the revised manuscript. The revised manuscript can be improved with further revisions to the text. My suggestions follow.

- (1) Authors should carefully check references and make sure that primary literature is properly cited. For example, the authors reference a review by Chase and Koelle instead of the primary literature reporting discovery of monoamine biosynthesis genes.
- (2) Lines 76-78 of the introduction are very confusing and should be rewritten.
- (3) In reference to Figure EV4, the authors should more clearly acknowledge that there are effects of mutant E. coli on respiration (OCR) that are not mediated by monoamine signaling and the pathway that they propose.
- (4) Temper use of 'systematically' and 'obviously' throughout the manuscript when describing approaches and results.
- (5) The discussion section does not need to summarize results and can instead focus entirely on discussion of the proposed pathway and its potential significance.
- (6) The manuscript should be checked throughout for minor grammatical/usage errors.

Referee #2:

The authors have done a nice job in addressing all of my comments. This is an interesting piece of work. Congratulations!

Comments and suggestions from Referee #1:

The revised manuscript of Li, Tong and colleagues includes substantial amounts of new data that address comments raised during review of the initial submission.

Important controls and cross-checks of key conclusions are included in the revised manuscript. The revised manuscript can be improved with further revisions to the text. My suggestions follow.

Response: We appreciate the reviewer for their positive evaluation of our revised manuscript and for their insightful suggestions to enhance the text. All comments have been carefully addressed, as outlined below.

(1) Authors should carefully check references and make sure that primary literature is properly cited. For example, the authors reference a review by Chase and Koelle instead of the primary literature reporting discovery of monoamine biosynthesis genes.

Response: We thank the reviewer for the constructive suggestion. In response, we have replaced the review article citation (Chase and Koelle, 2007) with the following four original research articles:

1. Hare EE, Loer CM (2004) Function and evolution of the serotonin-synthetic *bas-1* gene and other aromatic amino acid decarboxylase genes in *Caenorhabditis*. *BMC Evol Biol* 4: 24
2. Lints R, Emmons SW (1999) Patterning of dopaminergic neurotransmitter identity among *Caenorhabditis elegans* ray sensory neurons by a TGFbeta family signaling pathway and a Hox gene. *Development* 126: 5819-31
3. Sze JY, Victor M, Loer C, Shi Y, Ruvkun G (2000) Food and metabolic signalling defects in a *Caenorhabditis elegans* serotonin-synthesis mutant. *Nature* 403: 560-4
4. Duerr JS, Frisby DL, Gaskin J, Duke A, Asermely K, Huddleston D, Eiden LE, Rand JB (1999) The *cat-1* gene of *Caenorhabditis elegans* encodes a vesicular monoamine transporter required for specific monoamine-dependent behaviors. *J*

In addition, we have thoroughly reviewed the reference list throughout the manuscript to ensure that primary literature is appropriately cited.

(2) Lines 76-78 of the introduction are very confusing and should be rewritten.

Response: We thank the reviewer for their constructive suggestion. We have rewritten the text to enhance clarity in the revised manuscript (lines 76–78).

(3) In reference to Figure EV4, the authors should more clearly acknowledge that there are effects of mutant *E. coli* on respiration (OCR) that are not mediated by monoamine signaling and the pathway that they propose.

Response: We thank the reviewer for this valuable suggestion. In response, we have added the following description to the Results section in the revised manuscript (lines 379–381): “Notably, the partial restoration of OCR in wild-type worms treated with *ΔcyoB E. coli* following serotonin supplementation (Fig. EV4C) supports the existence of a monoamine signaling-independent pathway regulating mitochondrial function.”

(4) Temper use of 'systematically' and 'obviously' throughout the manuscript when describing approaches and results.

Response: We thank the reviewer for this valuable suggestion. We have carefully reviewed the entire manuscript and have replaced or removed the words ‘systematically’, ‘obviously’, and similar expressions to improve the clarity and objectivity of our descriptions.

(5) The discussion section does not need to summarize results and can instead focus entirely on discussion of the proposed pathway and its potential significance.

Response: We thank the reviewer for the constructive suggestion. In response, we have

revised the Discussion section by removing the extensive summary of results, thereby refocusing the discussion on the interpretation and broader implications of our findings.

(6) The manuscript should be checked throughout for minor grammatical/usage errors.

Response: We thank the reviewer for this valuable suggestion. We have carefully proofread the entire manuscript line by line to correct grammatical, typographical, and stylistic errors. Additionally, the manuscript has been edited by a professional English editing service to ensure the highest standard of language quality.

Referee #2:

The authors have done a nice job in addressing all of my comments. This is an interesting piece of work. Congratulations!

Response: We thank Referee #2 for their constructive, insightful, and highly supportive feedback throughout the revision process.

Dear Hongyun,

Thank you for incorporating the final changes in the manuscript. I am now pleased to inform you that your manuscript has been accepted for publication.

Before we forward your manuscript to the publishers, I noticed that some modifications in the synopsis image would be needed. Please correct "intestine cell" to "intestinal cell". Please also increase the font size and consider improving the resolution, as the text and the image become difficult to discern when the image is reformatted to 550 pixels in width according to our technical requirements.

I will also look into the synopsis text that you kindly provided and will let you know at the beginning of the next week if any edits to the journal style are needed.

If you have any questions, please do not hesitate to contact the Editorial Office or me directly. Thank you for your contribution to The EMBO Journal and congratulations with a nice study!

With best wishes,

Ieva
